# Metabolic regulation of proteome stability via N-terminal acetylation controls male germline stem cell differentiation and reproduction

Charlotte M. François [1], Thomas Pihl [1], Marion Dunoyer de Segonzac[1], Chloé Hérault [1] & Bruno Hudry [1] ✉

The molecular mechanisms connecting cellular metabolism with differentiation remain poorly understood. Here, we find that metabolic signals contribute to stem cell differentiation and germline homeostasis during *Drosophila melanogaster* spermatogenesis. We discovered that external citrate, originating outside the gonad, fuels the production of Acetyl-coenzyme A by germline ATP-citrate lyase (dACLY). We show that this pathway is essential during the final spermatogenic stages, where a high Acetyl-coenzyme A level promotes NatB-dependent N-terminal protein acetylation. Using genetic and biochemical experiments, we establish that N-terminal acetylation shields key target proteins, essential for spermatid differentiation, from proteasomal degradation by the ubiquitin ligase dUBR1. Our work uncovers crosstalk between metabolism and proteome stability that is mediated via protein post-translational modification. We propose that this system coordinates the metabolic state of the organism with gamete production. More broadly, modulation of proteome turnover by circulating metabolites may be a conserved regulatory mechanism to control cell functions.

Successful execution of the germline differentiation program requires intrinsic signals as well as paracrine and endocrine signalling among germ cells, supporting somatic cells, and other organs[1,2]. In the testes, many growth factors produced by somatic cells influence germ cell development by direct contact or by indirect, ligand-mediated signalling. In addition to the known peptide signals mediating germline-soma communication, products of intracellular metabolic pathways are also detected in the circulatory system. Recent work has revealed that tissues can use and sometimes require such exogenous, circulating metabolites for function[3–6]. For example, by examining the fluxes of circulating metabolites in mice, Hui and colleagues[4] established that the contribution of glucose to intrinsic tricarboxylic acid (TCA) metabolism is primarily indirect, via circulating lactate, in all tissues except the brain. There is therefore growing interest in exploring the

possible action of metabolites on tissue growth and homeostasis. In the context of germline development[7,8], outside of a known role for energy metabolism in germ cell differentiation, the significance of metabolic pathways remains largely unknown[8], this question is of particular interest given that fertility is affected by nutrition and the availability of energy reserves in most animal species[9–12].

We recently discovered sex differences in intestinal carbohydrate metabolism, which are extrinsically controlled by the adjacent testis, and govern food intake through gut-derived citrate[13]. During the course of these experiments, we also noticed that this inter-organ communication impacts sperm production, suggesting that citrate from the extracellular milieu might contribute to gametogenesis[13]. One outstanding question is to determine if the role of gut-derived citrate in sustaining sperm production is independent of its role in

[1]Université Côte d'Azur, CNRS, Inserm, Institut de Biologie Valrose, Nice 06108, France. ✉e-mail: Bruno.HUDRY@univ-cotedazur.fr

stimulating appetite. Especially if circulating citrate is imported into the germline cells and regulates directly spermatogenesis. Although mitochondrial production of citrate seems to be the primary source for most cells, citrate plasma concentrations are relatively high[14]. The functional importance of exogenous citrate transport by cells is still unclear and has therefore gathered increasing interest. For example, up-regulation of the citrate transporter expression has been reported in patients with non-alcoholic fatty liver disease. Accordingly, knockdown of the citrate transporter encoding gene (*Slc13a5*) prevents diet-induced non-alcoholic fatty liver disease in mice[15,16]. *Slc13a5*-knockout mice have also increased hepatic mitochondrial biogenesis, higher lipid oxidation, and energy expenditure, which protect the mice from obesity and insulin resistance[17]. Finally, citrate import is particularly important in human and mouse new-borns; mutations in *Slc13a5* causing SLC13A5-epilepsy, a type of early-onset epileptic encephalopathy[18,19]. Yet, the exact molecular mechanisms by which the functional deficiency of *Slc13a5*, and circulating citrate, influence these diverse phenotypes remain to be fully elucidated.

Here, we use *Drosophila melanogaster* spermatogenesis to investigate how a metabolic signal from the extracellular environment - citrate - instructs germline differentiation and homeostasis. We show that a protein post-translational modification couples the male metabolic state to the dynamics of germline stem cell differentiation. This original type of regulatory mechanism, mediated by circulating citrate and NatB-dependent N-terminal protein acetylation, controls the stability of a portion of the male germline proteome essential for spermatid individualisation.

## Results

### Citrate import is essential for male germline differentiation but not as a bioenergetics source

Sex differences in intestinal citrate homeostasis[13], and effects of citrate on male gamete differentiation, prompted us to investigate the potential role of this metabolite in male gonads. Cellular citrate is found in two separate pools: a mitochondrial one, where citrate is a substrate in the TCA cycle, its oxidation fuelling ATP production; and a cytosolic one, where citrate is broken down by the ATP-citrate lyase (dACLY, in flies also known as ATPCL) to produce Acetyl-coenzyme A (Acetyl-CoA) and oxaloacetate (OAA)[20]. Our initial hypothesis was that the requirement for citrate reflected the need for energy during spermatogenesis, and we therefore expected that citrate was required for TCA metabolism in the mitochondria.

To test this idea, we used the male fertility rate as a physiological readout for a genetic screen. We knocked down metabolic genes in the male germline, reasoning that the inhibition of citrate metabolising enzymes would reduce the male fertility rate if the metabolic pathway corresponding to this enzyme is essential for sperm production. Surprisingly, knocking down the 32 genes coding for the TCA cycle enzymes, using 44 different short hairpin RNA (shRNA) and RNA interference (RNAi) lines, did not affect male fertility (Fig. 1a and Table S1). As a positive control of our screen, we identified *Succinyl-coenzyme A synthetase β subunit, ADP-forming* (*ScsβA*) as the only TCA cycle gene essential for spermatogenesis. This TCA cycle component is known to carry a crucial moonlighting function in the male germline, having a structural role in spermatids, that can be uncoupled from its metabolic function[21]. These results being unexpected, we functionally validated the efficacy of our knockdowns, via ubiquitous expression in the soma (Fig. S1a). Importantly, all the tested lines, targeting enzymes for all the eight steps of the TCA cycle, efficiently knocked down expression of the corresponding enzymes, inducing lethality, as anticipated for genes coding for essential TCA cycle enzymes.

To further validate that the TCA cycle is dispensable for germline differentiation, we perform additional loss-of-function experiments, focusing on the first step of the TCA cycle, in which Acetyl-CoA condensates with OAA to form citrate. The fly genome encodes for two

citrate synthase genes: *dCS* and *CG14740*[22]. While *dCS* is ubiquitous, *CG14740* is expressed exclusively in the male germline[23]. Immunohistochemical analyses confirmed that this enzyme localises to the spermatid mitochondria (Fig. S1b). We engineered, by CRISPR-Cas9, a null mutant for this testis-specific citrate synthase. Our allele fully abolished *CG14740* expression (Fig. S1c), yet male fertility was unaffected (Fig. 1b). We also performed double knockdown experiments to rule out potential redundancy between the two citrate synthases, using validated effective RNAi lines (Fig. S1d). Males remained fertile (Fig. 1c), further confirming that the TCA cycle is dispensable for the production of sperm.

We therefore turned to the second way in which citrate is utilised, a pathway involving the dACLY enzyme and leading to cytosolic Acetyl-CoA production. Genetic manipulations, interfering with this route of citrate consumption via knockdown of *dACLY* (Fig. S1e), lead to male sterility (Fig. 1d). This finding shows that citrate is a key metabolite in the male germline and is used as fuel for the production of Acetyl-CoA.

To explore the origin of the citrate used by dACLY, we employed a genetic approach. From our previous results, we anticipated that cell-intrinsic production would not be a significant source of citrate. We indeed found that depleting the mitochondrial citrate transporter (*dCIC*, in flies known as *scheggia* (*sea*))[24,25] failed to affect male fertility (Fig. 1e, f). Using a genetically encoded GFP-based citrate sensor[26] expressed in the male germline, we verified that *dCIC/sea* or citrate synthase knockdowns did not impact the cytosolic citrate level (Fig. S1f, g). These results confirmed that germline mitochondria were not the source of the essential cytosolic citrate and that an alternative mechanism was involved. To investigate external import, we silenced by RNAi the three predicted citrate transporters (Fig. S1h), exclusively expressed in the male germline: *CG7309*, *CG33934*, and *I'm not dead yet 2* (*Indy-2*)[22,23]. Single downregulation of citrate transporter failed to affect male fertility (Figs. 1e, g, and S1i). However, while remaining fertile, males with all double and triple loss-of-function manipulations displayed a fertility rate divided by a factor of two and three, respectively (Fig. 1g). In contrast, parallel ectopic over-expression of one specific citrate transporter rescued male fertility to amounts comparable to those detected in wild-type males (Fig. 1g). These results indicate that male fertility depends on the extracellular import of citrate. Furthermore, the level of citrate entering in the male germline correlates with the degree of male fertility.

Altogether, these findings show that male germline differentiation relies on cytosolic citrate as an Acetyl-CoA precursor, but not as a bioenergetics source. Furthermore, while cell-autonomous citrate production in the germinal mitochondria is dispensable, the transport of germline-extrinsic citrate plays a critical role in sperm production.

### Metabolic signalling through the conversion of citrate into Acetyl-CoA plays a key role in spermatid differentiation

To determine the cause of the male sterility induced by *dACLY* downregulation, we examined the process of sperm differentiation. In flies, the testes exhibit a tubular organisation with a continuum of differentiating sperm cells arranged chronologically, allowing the simultaneous observation of all spermatogenic stages[27–29]. These stages include stem cell daughter cells called gonialblasts, which undergo transit-amplifying divisions and produce interconnected spermatogonial precursors. These spermatogonia then differentiate into spermatocytes and grow dramatically in size, before dividing twice by meiosis. The final stage, spermiogenesis, results in haploid elongated spermatids interconnected in a syncytium. The production of single motile sperm requires the encapsulation of each spermatid by an independent plasma membrane and the elimination of most sperm cytoplasm. This apoptosis-like process, known as sperm individualisation, relies on caspase activity[30,31] and is characterised by membrane-enclosed structures termed waste bags where the discarded cytoplasm accumulates,. The spermatozoa are then released from the testes into

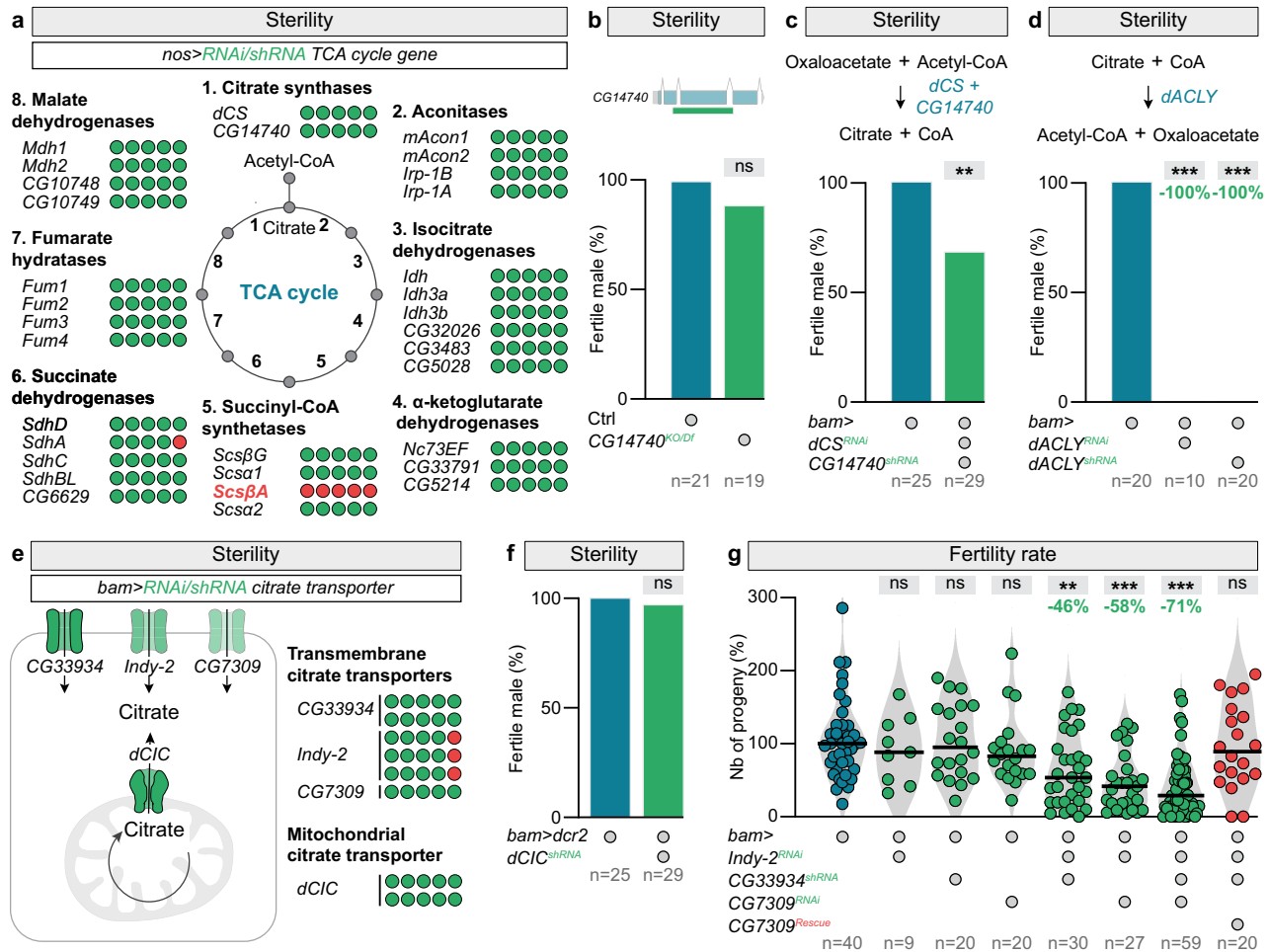

**Fig. 1 | Citrate import is essential for male germline differentiation but not as a bioenergetics source. a** The number of fertile (green circles) and sterile (red circles) males, expressing one RNAi targeting a specific TCA cycle gene under the control of the *nanos-Gal4* (*nos-Gal4*) driver (active in the male germline). **b** Fertility of *CG14740* knock-out males. On the top, a representation of the *CG14740* locus with the deleted region symbolised by a green line. **c** Percentage of fertile males expressing two RNAis targeting *citrate synthase* (*dCS*) and *CG14740* under the control of the *bag of marbles-Gal4* (*bam-Gal4*) driver (active in the germline). *p*-value from one-sided Mann–Whitney test is \*\**p* = 0.0066. **d** Percentage of fertile males expressing a RNAi targeting the *ATP citrate lyase* gene (*dACLY*) under the control of *bam-Gal4*. **e, f** The number of fertile (green circles) and sterile (red circles) males, expressing **e** a specific RNAi targeting a citrate transporter, or **f** *mitochondrial citrate carrier* (*dCIC*, in flies known as *scheggia* (*sea*)) under the control of the *bam-Gal4* driver. **g** The fertility rate of males with single, double, and triple transmembrane citrate transporter knockdowns, specifically in the germline. **d, g** *p* values from one-sided Krustal–Wallis tests are \*\**p* = 0.0073; \*\*\**p* < 0.0001. In all panels, *n* = number of males tested. In this and all subsequent figures, control datasets are displayed in blue, data related to loss-of-function experiments in green and results related to rescue experiments in red, unless otherwise indicated.

the seminal vesicles (SVs), where the sperm matures under the influence of seminal fluid and becomes motile.

To examine whether a decrease in cytosolic Acetyl-CoA levels disrupted this process, we reduced cytosolic Acetyl-CoA production in testes by germline-specific *dACLY* knockdown, and then characterised the impact on spermatogenesis using twenty-five protein reporters[32–39], expressed in various cellular components and stages of sperm differentiation. We could not detect any effect on mitotic spermatogonia, spermatocyte growth, nor spermatid elongation (Figs. 2a and S2a). However, our immunohistochemical analyses revealed that only the last step of sperm differentiation, the individualisation, was massively compromised (Fig. 2a). Indeed, motile sperm in the seminal vesicles (SVs, dashed lines in the Dj^GFP marker) and waste bags with an activated version of the caspase-3-like effector caspase, cleaved Death caspase-1 (cDcp-1) (WBs, asterisks in the cDcp-1 staining) at the end of the elongated axonemes were missing (Fig. 2a).

To confirm that the phenotypes were specific to the loss of dACLY, we investigated whether the shRNA-induced phenotypes could be reversed by re-expression of *dACLY*. We generated a synthetic shRNA-resistant transgene for this genetic complementation test,

which bears minimal sequence identity with the shRNA but encodes a wild-type dACLY protein. Expression of this shRNA-refractory construct rescued all the previously identified defects, including male fertility (Fig. 2b, c), mature sperm presence in the SVs (Fig. 2d, e), and waste bag formation (Fig. 2e, f).

To determine whether the defects induced by *dACLY* loss were confined to the final stages of sperm differentiation, we investigated two events, happening, just before individualisation: post-meiotic transcription and histone-to-protamine transition. In fly, post-meiotic transcription occurs just before protamines can be detected in spermatid nuclei[33], and is restricted to twenty comet and cup genes[40]. Germline-specific *dACLY* knockdowns did not affect *comet* and *cup* gene transcription (Fig. S2b) nor protamine incorporation (Fig. 2a). Thus, loss of dACLY affects individualisation itself, not the steps leading up to it.

We then sought to define the temporal window of *dACLY* requirement during the differentiation of the male germline stem cell lineage. To do so, we uncoupled the induction of *dACLY* knockdown from the re-expression of the shRNA-immune rescue construct. During spermatogenesis, there are three main stages of transcriptional

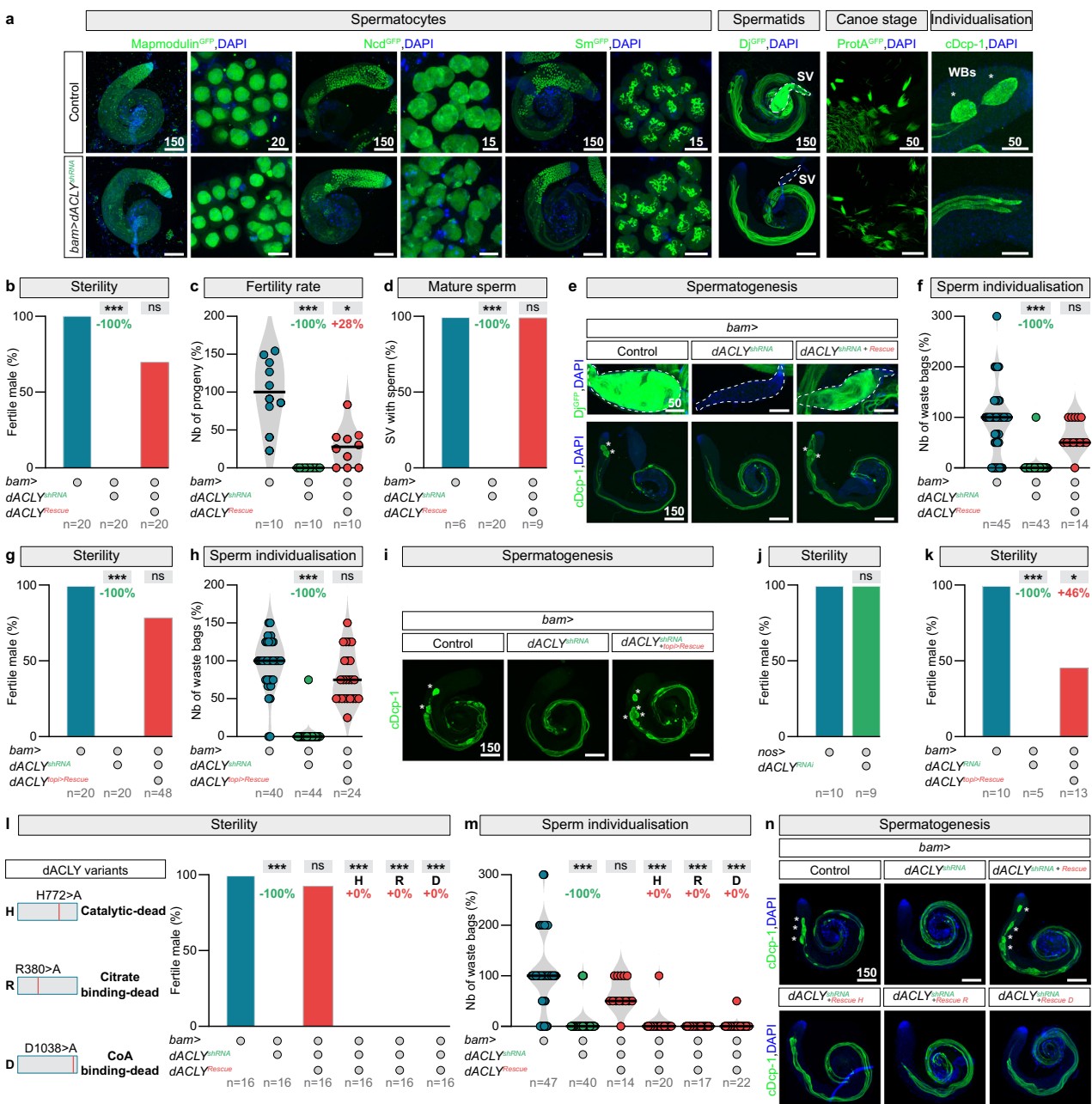

**Fig. 2 | Metabolic signalling through the conversion of citrate into Acetyl-CoA plays a key role in spermatid individualisation. a** Representative images (DNA: DAPI, blue; protein, green) of Mapmodulin$^{GFP}$, Non-claret disjunction (Ncd$^{GFP}$), Smooth (Sm$^{GFP}$), Don juan (Dj$^{GFP}$), Protamine A (ProtA$^{GFP}$), and cleaved Dead caspase-1 (cDcp-1) expressions in testes of control males and in males after germline-specific *dACLY* knockdown using *bam-Gal4*. Seminal vesicles (SVs) and waste bags (WBs) are indicated by dashed lines and asterisks respectively. **b**–**f** Quantifications of **b** the percentage of fertile males, **c** the number of progenies reaching pupal stage, **d** the percentage of seminal vesicles with mature sperm, **e** representative images (DNA: DAPI, blue; protein, green) of Dj$^{GFP}$, and cDcp-1 expressions, and **f** the number of waste bags in testes of control males, males with germline-specific *dACLY* knockdown using *bam-Gal4* and males with rescued *dACLY* knockdown using a *UAS-dACLY* transgene. **g**–**i** Quantifications of **g** the percentage of fertile males, **h** the number of waste bags, and **i** representative images (DNA: DAPI, blue; protein, green) of cDcp-1 expression in testes of control males, males

with germline-specific *dACLY* knockdown using *bam-Gal4* and males with rescued *dACLY* knockdown using a *topi > dACLY* transgene. **j** Percentage of fertile males expressing an RNAi targeting *dACLY* under the control of the *nos-Gal4* driver (active in the germline stem cells). **k** Quantifications of the percentage of fertile males in testes of control males, males with germline-specific *dACLY* knockdown using *bam-Gal4* combined with a different *dACLY* RNAi, and males with rescued *dACLY* knockdown using a *topi > dACLY* transgene. **l**–**n** Quantifications of **l** the percentage of fertile males, **m** the number of waste bags, and **n** representative images (DNA: DAPI, blue; cDcp-1, green) of cDcp-1 expression in testes of control males, males with germline-specific *dACLY* knockdown using *bam-Gal4*, and males with rescued *dACLY* knockdown using different mutated *UAS-dACLY* transgenes. Scale bars: in µm. *n* = number of flies tested in **b**, **c**, **g**, **j**, **k**, **l**, the number of testes analysed per genotype in **d**, **f**, **h**, and **m**. In all panel, *p* values from one-sided Krustal–Wallis tests are ***$p$ < 0.0001, in **C** *$p$ = 0.0219 and in **k** *$p$ = 0.0286.

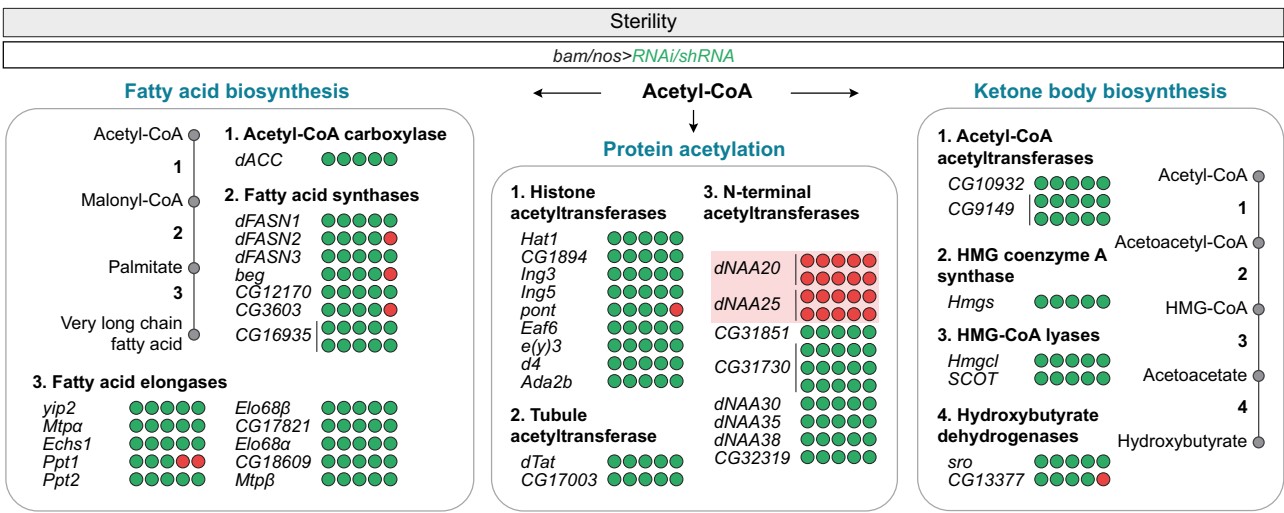

**Fig. 3 | N-terminal protein acetylation is essential for sperm production.** The number of fertile (green circles) and sterile (red circles) males, expressing an RNAi targeting one specific gene coding for an Acetyl-CoA utilising enzyme under the control of the germline-specific *bam-Gal4* or *nos-Gal4* drivers.

activity: one in stem cells, another in spermatogonia, and the last one in spermatocytes[27,29]. Toward the end of the spermatocyte stage, transcriptional activity is arrested and the chromatin is condensed for meiotic divisions. We reduced *dACLY* expression using a genetic tool active in spermatogonia (*bam-Gal4*[41]) driving *dACLY* shRNA (Fig. 2g–i) and at the same time delivered our rescue transgene in late spermatocytes (using the promoter of *matotopetli, topi*[42]), just before the transcriptional arrest. In this experimental setup, *dACLY* loss-of-function phenotypes were not observed and male fertility was restored (Fig. 2g–i). In parallel, stem cell-specific expression of *dACLY* RNAi (using the *nanos-Gal4*[43]) did not cause male sterility (Fig. 2j). We obtained similar results with another RNAi line targeting a different exon of *dACLY*. Spermatogonial-specific expression of this RNAi caused male sterility, and was rescued by re-expression of *dACLY* in late spermatocytes only (Fig. 2k). These data establish that dACLY is dispensable during the early steps of spermatogenesis, but necessary during the last step, consistent with the individualisation defects observed.

We then investigated whether the enzymatic activity of dACLY is essential for male fertility. We generated three dACLY variants presenting mutations in the catalytic site (H772 > A)[44,45], and in the two substrates binding domains (citrate binding domain (R379A)[44], and the CoA binding pocket (D1038A)[44]) (Fig. 2l), and performed a rescue-based structure-function analysis. The three mutant versions of dACLY were properly expressed in testes (Fig. S2c), but all failed to rescue the *dACLY* loss-of-function phenotypes (Fig. 2l–n). These results support the idea that the essential function of dACLY during male germline differentiation is mediated by its Acetyl-CoA synthetase catalytic activity.

Acetyl-CoA is a key player with diverse roles in metabolism. Besides citrate, it can also derive from other precursors, such as, acetate, fatty acids, and amino acids[46,47]. We therefore wondered whether other Acetyl-CoA biosynthesis pathways may compensate, fully or partially, for *dACLY* inactivation. None of the other enzymes, generating Acetyl-CoA, drove male sterility upon germline-specific inactivation (Fig. S2d), nor were they transcriptionally up-regulated following *dACLY* knockdown (Fig. S2e). These results indicate that the different routes of Acetyl-CoA production act independently in the male germline and the alternative pathways do not normally compensate for loss of the main dACLY-dependent one. Possible explanations for this lack of compensation between different pools of Acetyl-CoA could be due to subcellular compartmentalisation and to the relative expression level of these Ac-CoA-producing enzymes.

Interestingly, dACLY, among the cytosolic enzymes, has the highest expression level[48].

Together, our data reveal a metabolic regulatory pathway in which circulating citrate, which is imported into the male germline by citrate transporters, is cleaved by dACLY into Acetyl-CoA. This dACLY-dependent cytosolic Acetyl-CoA production is essential to promote spermatid individualisation.

### N-terminal protein acetylation is essential for sperm production

We then explored the Acetyl-CoA potential biological roles during spermatid individualisation. The accumulation of Acetyl-CoA in the cytoplasm is known to fuel lipid synthesis and histone acetylation[49–57]. Both processes depend on dACLY activity in many cell types[58–65]. We hypothesised that the depletion of key enzymes using Acetyl-CoA should mimic *dACLY* sterility phenotype if the targeted pathway is the main route of Acetyl-CoA consumption. To test this hypothesis, we knocked down 25 genes involved in fatty acid (FA) synthesis and 19 acetyltransferases with enriched expression in adult testes (Table S2).

Very interestingly, among all the 44 tested genes encoding Acetyl-CoA utilising enzyme, only the silencing of two genes (Fig. S3a, b), belonging to the same NatB complex, *dNAA20* and *dNAA25* (in fly also called *psidin*), induced male sterility (Fig. 3). By contrast, knocking down genes coding for the FA biosynthesis enzymes did not impact male fertility (Fig. 3). Previous research[66–68] suggested that fatty acid supplementation could partially rescue the phenotype of ubiquitous *dACLY* mutation at the larval stages. In the male germline, fatty acid supplementation did not restore sperm individualisation (Fig. S3c, f) or mature sperm presence in the seminal vesicles (Fig. S3d, f). Even though males were given fatty acids, they remained sterile (Fig. S3e), indicating that the observed defects following *dACLY*-dependent cytosolic Acetyl-CoA production inhibition are not due to a decrease in the supply of fatty acids. RNAi-mediated silencing of 9 histone acetyltransferases (HATs), and 2 α-tubulin acetyltransferases (TATs) all failed to affect male fertility (Fig. 3), suggesting that these enzymes were unlikely the key players using Acetyl-CoA. These results are in agreement with our previous findings. Indeed, the histone-to-protamine transition requires a massive histone hyper-acetylation (on H4 and H3)[69,70] and this process is not affected by *dACLY* loss (Fig. 2a). Furthermore, in lysates of dissected testes acetylation of histone H3 was not reduced upon *dACLY* silencing (Fig. S3g). Under some conditions, Acetyl-CoA can also be used in the biosynthesis of ketone bodies, acetoacetate, and β-hydroxybutyrate[46]. But again,

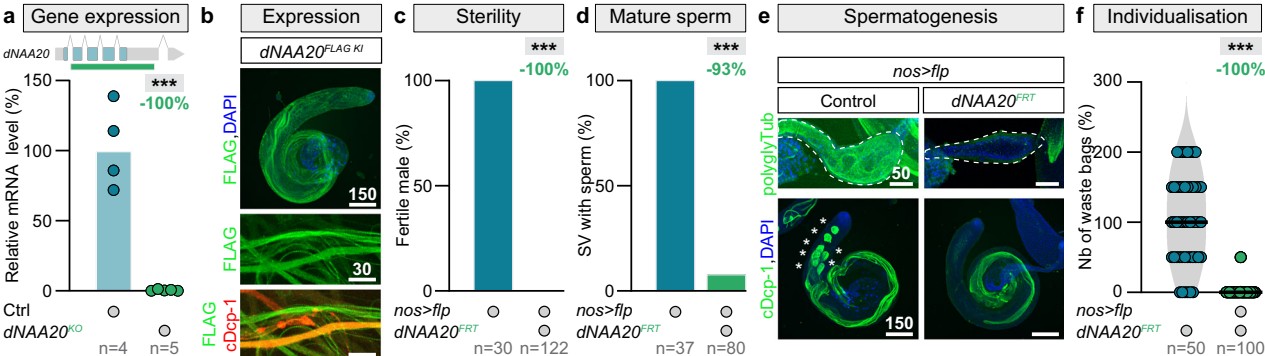

**Fig. 4 | dNAA20 is essential for male germline differentiation. a** RT-qPCR expression data for *dNAA20* in whole control larvae and *dNAA20* knock-out individuals. On the top, a representation of the *dNAA20* locus with the deleted region symbolised by a green line. *p* value from one-sided *t* test is ***$p$ < 0.0001. **b** Representative images (DNA: DAPI, blue; dNAA20: FLAG, green, individualising spermatids: cleaved Dead caspase-1 (cDcp-1), red) of dNAA20$^{FLAG}$ knock-in testicular expression. **c**–**f** Quantifications of **c** the percentage of fertile males, **d** the percentage of seminal vesicles with mature sperm, **e** representative images (DNA: DAPI, blue; protein, green) of polyglycylated α-tubulin (polyglyTub), and cDcp-1 expressions, and **f** the number of waste bags in testes of control males, and males with germline-specific *dNAA20* knock-out. Scale bars: in μm. *n* = number of biological replicates analysed, each replicates containing 10 larvae in **a**, the number of flies tested in **c**, the number of seminal vesicles analysed in **d** and number of testes analysed per genotype in **f**. In all panel, *p* values from one-sided Mann–Whitney tests are ***$p$ < 0.0001.

knocking down the 7 enzymes involved in ketogenesis did not induce male sterility (Fig. 3).

Testis-enriched paralogs of more broadly-expressed genes are common[71–74]. Interestingly, two duplicate genes, *CG31851* and *CG31730*, paralogs of *dNAA20*, are exclusively expressed in testes[22,23]. Although our RNAi screen did not identify *CG31851* and *CG31730* individually as essential genes for male germline differentiation (Fig. 3), we tested whether these paralogs could act in a redundant manner, as alternative NatB catalytic subunits. Testis-specific expression was confirmed by engineering HA-tagged knock-in for CG31851 followed by immuno-histochemical analyses (Fig. S3h). We then generated a deletion removing *CG31851* and *CG31730* by CRISPR-Cas9. This mutation fully abolished the expression of both genes (Fig. S3i) and males carrying this deletion remained fertile (Fig. S3j, k). No defect in spermatid individualisation (Fig. S3l), nor mature sperm formation was observed (Fig. S3m). Even though these two proteins are divergent from dNAA20, with 60% of their residues being different, we tested their ability to rescue *dNAA20* loss-of-function. Both proteins individually could not rescue *dNAA20* knockdown phenotype indicating that dNAA20 is the only essential NatB catalytic subunit active in the male germline (Fig. S3n).

These data suggest that imported citrate, cleaved by dACLY to produce cytosolic Acetyl-CoA, is consumed, at least partially, to support N-terminal (Nt) protein acetylation mediated by the NatB complex.

## N-terminal protein acetylation mediated by NatB is essential for spermatid differentiation

We therefore focused our attention on characterising NatB function during spermatogenesis. The NatB complex is formed by the catalytic subunit dNAA20 and the auxiliary subunit dNAA25[75,76]. It co-translationally acetylates N-termini starting with methionine, followed by an acidic residue (MD-, ME-) or their amide (MN-, MQ-)[77]. NatB substrates represent approximately 20% of the fly proteome and 90% of the proteins of this substrate class are irreversibly acetylated[77]. Although the majority of eukaryotic proteins are subjected to Nt acetylation[75,76], there is no specific evidence of Nt protein acetylation function during male germline differentiation. More broadly, the biological relevance of Nt protein acetylation in the context of animal development remains poorly understood.

To explore how Nt protein acetylation controls male gamete formation, we started by characterising *dNAA20* loss-of-function phenotypes. We engineered the first complete *dNAA20* null mutant in flies using CRISPR-Cas9. This mutant allele fully abolished *dNAA20*

expression (Fig. 4a) and was lethal. This lethality is rescued by ubiquitous re-expression of a wild-type form of dNAA20 (Fig. S4a). Since, *dNAA20* null mutant is lethal, we generated two different FRT-flanked knock-in alleles, with tagged or untagged rescue transgenes. The FLAG-tagged knock-in revealed that dNAA20 protein is ubiquitously expressed in the somatic tissues (Fig. S4b). In the male germline, dNAA20 is also expressed and co-localises with a marker of individualising spermatids (Fig. 4b, see zooms). Next, focusing on the function in the germline, we combined our new FRT-flanked knock-in alleles with germline-specific FRT recombinase (Flp) to produce germinal mutant cells. As observed with RNA silencing: males were sterile (Fig. 4c). Furthermore, mature sperm were absent in the SVs (Fig. 4d, e), and waste bag formation was abolished (Fig. 4e, f). All these results together strongly indicate that dNAA20 is essential for spermatid individualisation.

Using rescue experiments, we then determined the temporal requirement for NatB function and whether its enzymatic activity is essential for male fertility. Germline-specific NatB subunit (*dNAA20* and *dNAA25*) knockdowns revealed a specific impairment in spermatid individualisation (Fig. S5a, b), similar to the defects caused by *dACLY* knockdown and dNAA20 knock-out. Re-expression experiments using RNAi-resistant *dNAA20* and *dNAA25* transgenes rescued all the previously identified defects, including male fertility (Fig. 5a, b, e, f), mature sperm presence in the SVs (Fig. 5c, g, i), and waste bag formation (Fig. 5d, h, i). Furthermore, we found that the phenotypes caused by inactivation of *dNAA20* and *dNAA25* in spermatogonia could be reversed by specific expression of rescue transgenes in late spermatocytes (Fig. 5a–d, e–h, i and Fig. S5c, d). Finally, knockdown of NatB subunits via RNAi in early germline stem cells did not affect spermatogenesis or male fertility (Fig. 5j, k). Thus, like dACLY, NatB activity is required during late spermatogenesis for individualisation. We next made a series of four point mutations in dNAA20 (Fig. 5l) to determine if the Nt acetyltransferase activity of the NatB complex was necessary to drive wild-type germline differentiation. We mutated the catalytic[78–80], Acetyl-CoA binging[81,82], and dNAA20-dNAA25 interaction sites[78,79] (Fig. 5l). Disruption of these sites completely abolished the capacity of dNAA20 to support spermatid individualisation and male fertility (Fig. 5l–n).

Together, these experiments show that the N-acetyltransferase activity of NatB complex is necessary for spermatid individualisation, but is dispensable during the early steps of male germline differentiation. Our results reveal that Nt protein acetylation is physiologically essential for animal germline homeostasis.

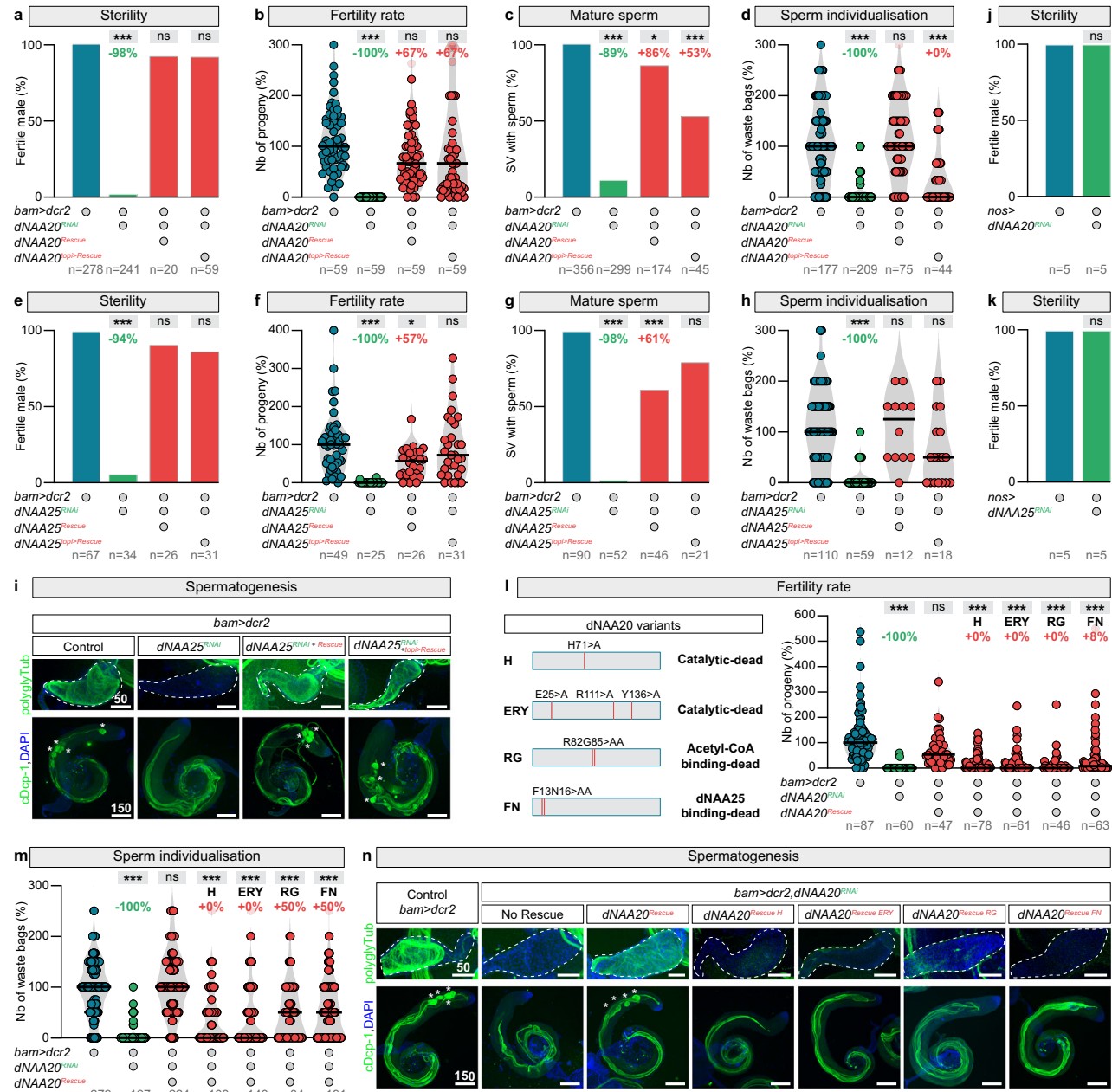

**Fig. 5 | NatB-mediated N-terminal protein acetylation is essential for spermatid individualisation. a–d** Quantifications of **a** the percentage of fertile males, **b** the number of progenies, **c** the percentage of seminal vesicles with mature sperm, and **d** the number of waste bags in testes of control males, males with germline-specific *dNAA20* knockdown using *bam-Gal4* and males with rescued *dNAA20* knockdown using a *UAS-dNAA20* or a *topi > dNAA20* transgene. **e–i** Quantifications of **e** the percentage of fertile males, **f** the number of progenies, **g** the percentage of seminal vesicles with mature sperm, **h** the number of waste bags, and **i** representative images (DNA: DAPI, blue; protein, green) of polyglycylated α-tubulin (polyglyTub), and cleaved Dead caspase-1 (cDcp-1) expressions, and in testes of control males, males with germline-specific *dNAA25* knockdown using *bam-Gal4* and males with rescued *dNAA25* knockdown using a *UAS-dNAA25* or a *topi > dNAA25* transgene.

**j** Fertility of control males and males with germline-specific *dNAA20* knockdown using *nos-Gal4* (active in the germline stem cells). **k** Fertility of control males and males with germline-specific *dNAA25* knockdown using *nos-Gal4*.
**l–n** Quantifications of **l** the number of progenies, **m** the number of waste bags and **n** representative images (DNA: DAPI, blue; protein, green) of polyglyTub, and cDcp-1 expressions in testes of control males, males with germline-specific *dNAA20* knockdown using *bam-Gal4* and males with rescued *dNAA20* knockdown using different mutated *UAS-dNAA20* transgenes. Scale bars: in μm. *n* = number of flies tested in **a**, **b**, **e**, **f**, **j**, **k**, and **l**; number of seminal vesicles analysed in **c**, and **g**, and number of testes analysed per genotype in **d**, **h**, and **m**. In all panel, *p* values from one-sided Krustal–Wallis tests are ****p* < 0.0001 in **c** **p* = 0.0413 and in **f** **p* = 0.0408.

## NatB-mediated N-terminal-acetylation blocks protein degradation

Next we investigated which of the processes regulated by Nt protein acetylation is critical for spermatid individualisation. Like other post-translational modifications, Nt protein acetylation can influence many aspects of protein functions, including protein-protein interactions, protein complex formation, protein subcellular targeting, or protein folding[75,76]. In addition, Nt protein acetylation was shown to play a role in the regulation of protein turnover through proteasomal degradation pathways[83–85]. Indeed, biochemistry analyses showed that N-terminally acetylated proteins are targeted for degradation via an Ac/N-end rule pathway[86,87], whereas unacetylated proteins are targeted by an Arg/N-end rule pathway[88,89]. However, more recent proteomic studies from yeast[90,91] and human cells[92] argue against the involvement

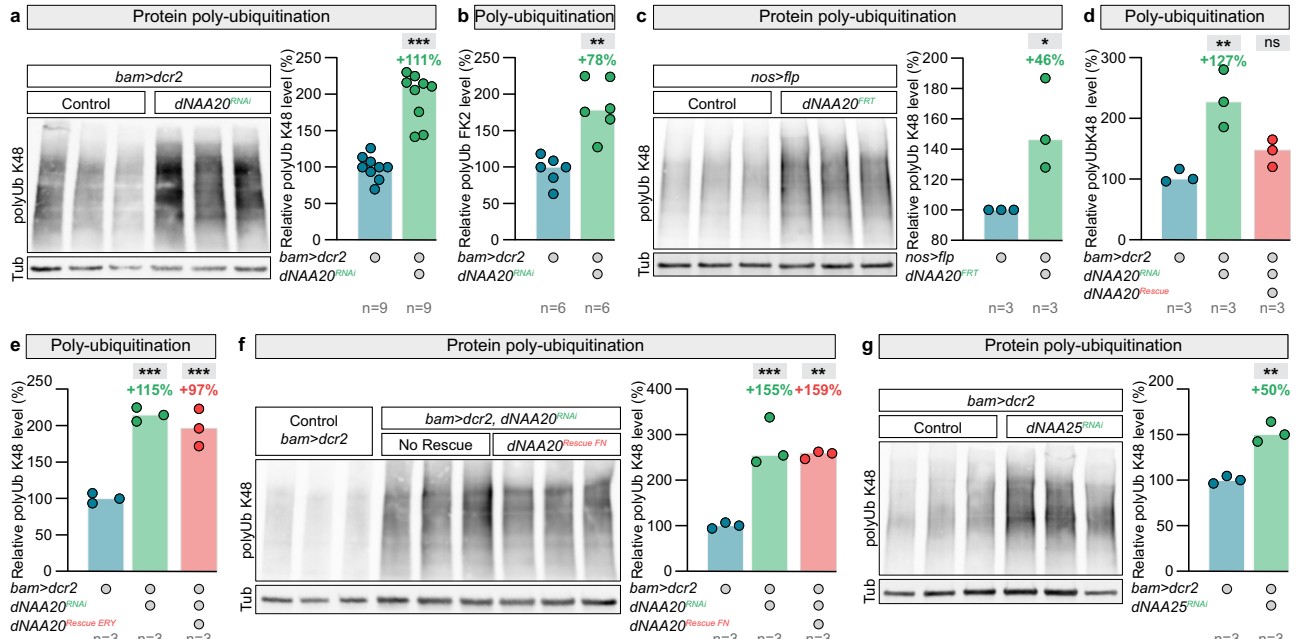

**Fig. 6 | NatB-mediated N-terminal-acetylation blocks protein poly-ubiquitination. a** Western blot analysis of lysine 48 (K48)-linked protein poly-ubiquitination in total extracts of dissected testes of control males and males with germline-specific *dNAA20* knockdown using *bam-Gal4*. Quantifications and a representative image of one blot are shown. **b** Quantifications of the level of protein ubiquitination using the FK2 antibody in total extracts of dissected testes of control males and males with germline-specific *dNAA20* knockdown using *bam-Gal4*. **a, b** *p* values from one-sided Mann–Whitney tests are **$p = 0.0022$ and ***$p < 0.0001$. **c** Western blot analysis of lysine 48 (K48)-linked protein poly-ubiquitination in total extracts of dissected testes of control males and males with germline-specific *dNAA20* knock-out. Quantifications and a representative image of one blot are shown. *p* value from one-sided *t* test is *$p < 0.0365$. **d–f** Quantifications of the level of K48-linked protein poly-ubiquitination in total extracts of dissected testes of control males, males with germline-specific *dNAA20* knockdown using *bam-Gal4*, and males with rescued *dNAA20* knockdown using (**d**) a wild type *UAS-dNAA20* transgene, **e** a catalytic dead, or **f** a dNAA25-binding dead *UAS-dNAA20* variants. **d–f** *p* values from one-ANOVA are ***$p < 0.0001$, **$p < 0.0044$. **g** Quantifications of the level of K48-linked protein poly-ubiquitination in total extracts of dissected testes of control males and males with germline-specific *dNAA25* knockdown using *bam-Gal4*. *p* value from one-sided *t* test is **$p = 0.0016$. *n* = number of biological replicates analysed, each replicates containing 150 testes.

of the N-end rule pathways in the regulation of proteome stability and suggest instead that Nt acetylation status is rarely used as a degradation signal. Furthermore, although NatB acetylates hundreds of different substrates in flies[77], nothing is known about the functional and physiological relevance of these acetylation events.

To assess the possible contribution of protein degradation in the context of spermatogenesis, we performed immunoblot analyses using an antibody specific for lysine 48 (K48)-linked poly-ubiquitinylated conjugates. K48-linked polyubiquitin chains are the most abundant and the canonical signals for protein degradation by the proteasome. We first compared the levels of ubiquitinated proteins in wild-type and *dNAA20* RNAi knockdown testes. Analysing lysates from dissected testes (Fig. 6a), we observed a dramatic increase in polyubiquitinylated proteins in knockdown male testes (Fig. 6a). Similar results were observed using the FK2 antibody[93] specific for mono- and polyubiquitinylated conjugates (Fig. 6b). Furthermore, the effect of dNAA20 RNAi on global protein turnover could also be recapitulated using the germline-specific *dNAA20* knockout (Fig. 6c). These results demonstrate an increase in protein degradation by the 26 S proteasome in the absence of NatB-mediated Nt acetylation, suggesting that Nt-protein acetylation may function to increase overall proteome stability.

To test if this effect of NatB loss-of-function is indeed driven by the direct lack of NatB-mediated Nt-acetylation we re-expressed wild-type and catalytically dead versions of *dNAA20*. While wild-type dNAA20 was capable of rescuing protein stability (Fig. 6d), as shown by the decrease in K48-linked polyubiquitinylated proteins, the catalytic dead dNAA20 mutant (ERY) was not (Fig. 6e). These results indicate

that Nt-acetylation is required for preventing an increase in K48-linked polyubiquitylation.

Finally, we found that this dNAA20 function is also dependent on dNAA25 since (1) expression of a dNAA20 variant unable to bind dNAA25 was not capable to promote target stability (Fig. 6f) and (2) germline-specific *dNAA25* RNA silencing similarly increased protein polyubiquitylation (Fig. 6g).

These data establish that NatB-mediated Nt-acetylation directly and positively promotes protein stability. Furthermore, they suggest that spermatogenesis offers a unique opportunity to investigate the physiological relevance, in animals, of proteome stability regulation by Nt protein acetylation.

## The N-recognin dUBR1 targets non-Nt acetylated NatB substrates for degradation
We tested if NatB function during spermatid individualisation is indeed mediated by the stabilisation of its targets. Non-Nt acetylated proteins are targeted for degradation by UBR-box ubiquitin E3 ligases, called N-recognins, which recognise non-Nt acetylated protein N-termini[94–99] and catalyse their K48-linked polyubiquitylation. The *Drosophila* genome encodes three N recognins (*dUBR1*, *dUBR4* also called *poe*, and *dUBR5* also called *hyd*). We reasoned that, if the *dNAA20* germline mutant phenotype is indeed mediated by increased protein degradation, inhibition the E3 enzymes targeting NatB substrates would increase male fertility. We therefore tested whether knockdown of *dUBR1*, *dUBR4*, or *dUBR5* could rescue male fertility in flies lacking *dNAA20* in the germline (Figs. 7a and S6a). Astonishingly, knockdown of the E3 ligase, *dUBR1* (Fig. S6b), rescued male fertility (Fig. 6a, b),

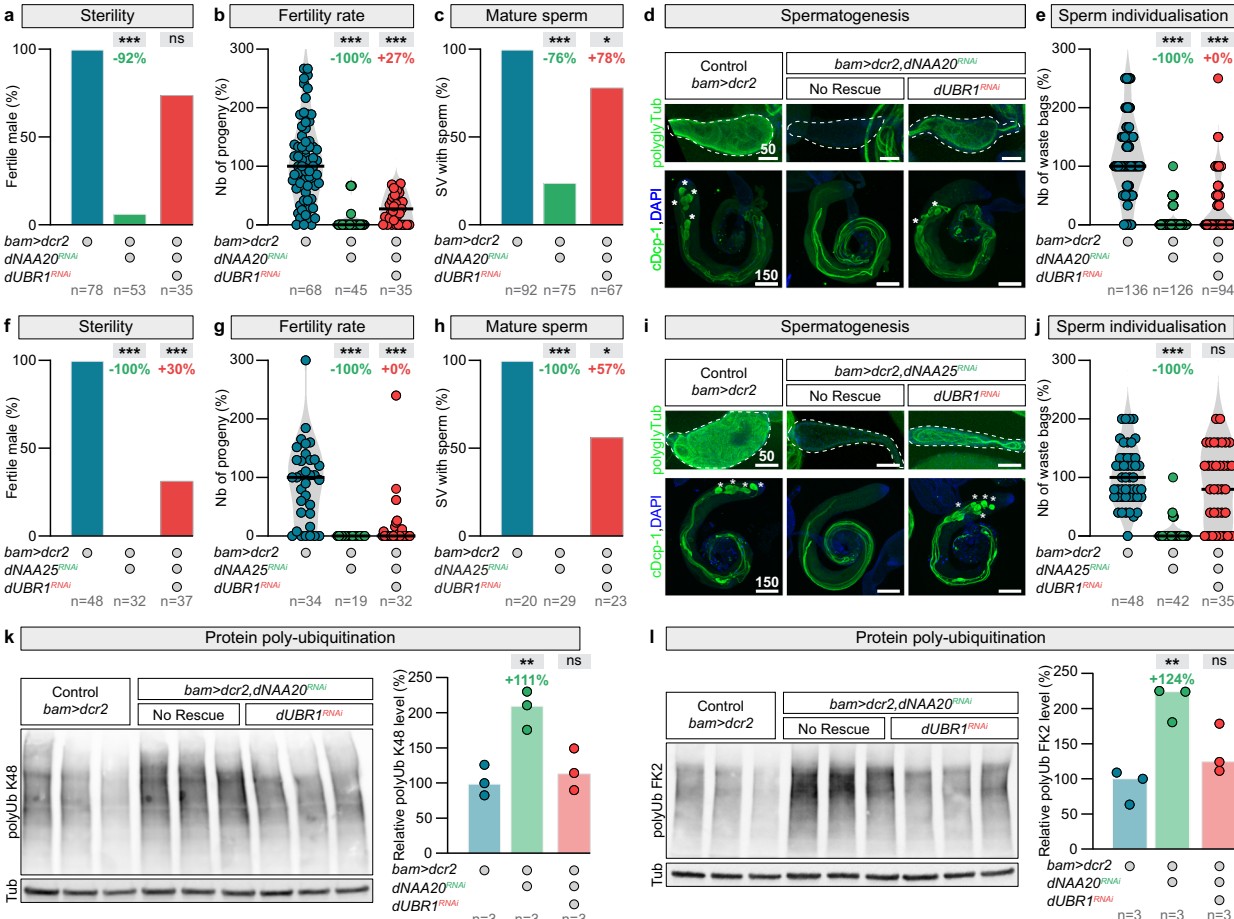

**Fig. 7 | The N-recognin dUBR1 targets non-Nt-acetylated NatB substrates for degradation. a–e** Quantifications of **a** the percentage of fertile males, **b** the number of progenies, **c** the percentage of seminal vesicles with mature sperm, **d** representative images (DNA: DAPI, blue; protein, green) of polyglycylated α-tubulin (polyglyTub), and cleaved Dead caspase-1 (cDcp-1) expressions, and **e** the number of waste bags in testes of control males, males with germline-specific *dNAA20* knockdown using *bam-Gal4* and males with double *dUBR1, dNAA20* knockdowns. Seminal vesicles and waste bags are indicated by dashed lines and asterisks respectively. **f–j** Quantifications of the percentage of **f** fertile males, **g** the number of progenies, **h** the percentage of seminal vesicles with mature sperm, **i** representative images (DNA: DAPI, blue; protein, green) of polyglyTub, and cDcp-1 expressions, **j** the number of waste bags in testes of control males, males with germline-specific *dNAA25* knockdown using *bam-Gal4* and males with double

*dUBR1, dNAA25* knockdowns. **k** Quantifications of the level of lysine 48-linked protein poly-ubiquitination in total extracts of dissected testes of control males, males with germline-specific *dNAA20* knockdown using *bam-Gal4*, and males with rescued *dNAA20* knockdown using *dUBR1 RNAi*. **l** Quantifications of the level of protein ubiquitination using the FK2 antibody in total extracts of dissected testes of control males, males with germline-specific *dNAA20* knockdown using *bam-Gal4*, and males with rescued *dNAA20* knockdown using *dUBR1 RNAi*. Scale bars: in μm. *n* = number of flies tested in **a**, **b**, **f**, and **g**; number of seminal vesicles analysed in **c**, **h**, number of testes analysed per genotype in **e**, **j**, and number of biological replicates analysed, each replicates containing 150 testes for **k** and **l**. In all panel, *p* values from one-sided Krustal–Wallis tests are ***$p < 0.0001$, in **c** *$p = 0.0156$ and in **h** *$p = 0.0178$. **k**, **l**, *p* values from one-sided Anova tests are **$p < 0.0058$.

mature sperm presence in the SVs (Fig. 7c, d), and to a lesser extent waste bag formation (Fig. 7e). *dUBR1* inactivation also rescued the loss-of-function phenotypes caused by loss of the auxiliary subunit *dNAA25* (Fig. 7f–j). Consistent with this rescue of male spermatogenesis at the cellular- and tissue-level, *dUBR1* inactivation also led to a reduction in K48-linked polyubiquitinylated proteins (Fig. 7k), and mono- and polyubiquitinylated conjugates (Fig. 7l). We used two GFP reporters to confirm that dUBR1-mediated K48 poly-ubiquitination triggers targets to the proteasome for degradation. Upon proteolytic cleavage, these markers expose either a stabilising amino acid (methionine, Met-GFP) or a destabilising amino acid targeted by the NatB complex (asparagine, Asn-GFP)[100]. The absence of methionine in the second reporter replicates a lack of N-terminal acetylation. The results demonstrated that the Met-GFP reporter protein produces a strong signal in the male germline (Fig. S6c), while the Asn-GFP marker is weak (Fig. S6c) and requires *dUBR1* knockdown for stabilisation (Fig. S6d). Combined together, these data indicate that the only

biological role of NatB-mediated Nt acetylation is shielding a key fraction of spermatid proteome from proteasomal degradation by the specific ubiquitin ligase, dUBR1. The data further suggest that Nt acetylation is a critical mechanism that couples the dynamics of sperm production with the male metabolic state.

Finally, we asked whether the regulatory pathway we discovered was active in other cellular contexts. We first asked whether this pathway is also required in the somatic gonad. We reduced the expression of all components of the pathway in the somatic part of the testes (using *traffic jam-Gal4* (*tj-Gal4*) to downregulated *dACLY*, *dNAA20*, and *dNAA25* expression in somatic gonadal cells[101]) (Fig. S6e). None of these genetic manipulations affected male fertility, indicating that activity of this metabolic signalling pathway in the somatic cells of the male gonad is dispensable for male fertility. We then asked whether this pathway may also explain the lethality caused by global loss of dNAA20. However, we found that inactivation of *dUBR1* in all the somatic tissues outside the gonad could not rescue the developmental

lethality caused by the global knockdown of *dNAA20* (Fig. S6f), suggesting that another molecular mechanism is also at play. Lastly, we asked whether this pathway may also regulate female fertility. We found that silencing of *dACLY*, *dNAA20*, or *dNAA25* in the female germline (Fig. S6g) had no effect on female fertility.

Altogether, our results indicate that Nt protein acetylation plays a critical role in controlling proteome stability and differentiation in a sex- and cell-type-specific manner.

## Discussion

Together, our data reveal a regulatory cascade whereby circulating citrate, which is imported into the male germline by citrate transporters, is cleaved by dACLY into Acetyl-CoA. The cytosolic Acetyl-CoA supports NatB-mediated Nt-protein acetylation. The imprinting of a fraction of the male germline proteome by acetylation controls the stability of key target proteins, essential for spermatid individualisation. This unique pathway of cellular proteostasis couples sperm production and male fertility with the internal metabolic state. At least two aspects of this metabolic regulation are remarkable: its specificity and the signals involved.

Previous work has provided conflicting evidence regarding the role of Nt acetylation in the regulation of protein stability. Our data show that this function of Nt acetylation in regulating protein degradation is stage-, cell type-, and sex-specific, being essential during late spermatogenesis, but dispensable in germline stem cells, the somatic part of the male sex organs and in the female germline. Moreover, the role of Nt acetylation during spermatogenesis is encoded exclusively by NatB. Indeed, NatA[102], NatC (Fig. 3), and NatE[103,104] catalytic subunits are dispensable in the male germline. Even in the somatic tissues outside the gonads, where NatB is essential during development, a different molecular mechanism appears to be in place. Thus, the function of Nat enzymes is cell and tissue-specific.

These results raise another question: how can a ubiquitous enzyme, like NatB, nevertheless control the differentiation of the male germline using a unique mechanism? The context-dependent function of NatB-mediated Nt protein acetylation could be determined by testis-specific factors. Interestingly, the three transmembrane citrate transporters, *Indy-2*, *CG7309*, and *CG33934*, are not only expressed exclusively in the male germline but are restricted to the late steps of spermatogenesis (Fig. S6h). dNAA20 expression is also enriched at these stages, suggesting that these restricted expression patterns could underlie the specific requirement of Nt protein acetylation for spermatid individuation during germline differentiation.

Our results, looking at the stability of a set of NatB targets (Fig. S6i), combined with the restricted phenotype of NatB loss-of-function, indicate that this proteostasis pathway likely controls a relatively small fraction of the proteome. The specific substrates of NatB in male germ cells remain to be identified and may well include testis-specific targets. Given that, at this time, Nt protein acetylation can only be detected by mass-spectrometry, cell type-specific analyses of differentiating fly male germline are challenging. Furthermore, full characterisation of this pathway may require further analysis of the degradative apparatus downstream of NatB. As with other UBR family members[105], the dUBR1 interaction with E2 ubiquitin-conjugating enzymes may depend on an additional specific E3 ubiquitin ligase factor, perhaps one of numerous uncharacterised E3 ligases presenting testis-specific expression[106,107]. A unique male germline-specific dUBR1-E3 protein complex could target the key unacetylated NatB substrates for degradation, and may provide another level of specificity.

Our data also provide further support for the idea that circulating citrate, a metabolite best known for its roles in mitochondria as a TCA cycle intermediate, can have systemic effects outside producing cells. Indeed, our experiments show clearly that the TCA cycle is dispensable for sperm differentiation in flies and that the germline depends on external citrate import for differentiation. Similarly, in mammals,

somatic Sertoli cells also act as a paracrine source of metabolites for the developing gametes[108,109]. Furthermore, citrate is found in human blood at high levels, but very little is known about its role as a signal across organs.

Citrate signalling may also exhibit context-dependent specificity. We have previously shown that carbohydrate handling is male-biased in a specific portion of the fly adult intestine and triggers citrate secretion specifically in males[13]. Interestingly, the reduction of gut-derived citrate production results in a significant decrease in individualising spermatids[13], mirroring, the phenotype obtained by NatB or citrate transporters (Fig. S6j) germline-specific loss-of-functions. But this manipulation does not sterilise males, suggesting that other organs contribute to the circulating citrate pool.

The biological significance of metabolite shuttling has been recognized since the discovery of the Cori cycle[110]. However, our findings that intestinal-produced citrate is taken up by testes cells extend this notion to another metabolite and organs. The instructive role of citrate in inter-organ communication may well extend to other biological contexts. Indeed, a recently-published atlas of inter-organ metabolite exchange in pigs, shows that the kidney specifically uptakes circulating TCA cycle intermediates, the most abundant of which is citrate[6]. Further work will be needed to determine the physiological role of this exchange.

Nt acetylation is one of the most common, but least understood post-translational modifications of eukaryotic proteins. In animals, its biological roles remain a mostly unexplored territory. In yeast, some NatB targets have been identified and characterised[111,112]. For example, NatB-mediated Nt acetylation of Tropomyosin1p (Tpm1p) is required for Tpm1p association with actin filaments[113,114]. However, until now, studies examining the role of acetylation during sperm production have been limited to internal lysine acetylation[115]. To our knowledge, our work provides the first evidence of the functional importance of Nt acetylation for spermatogenesis and reproduction.

Could the role of Nt protein acetylation during male gametogenesis be conserved in mammals? Interestingly, the human *hACLY* (Fig. S6l and S66k) and *hNAA20* (Fig. S6m–p) orthologs could rescue the loss of the corresponding fly proteins, suggesting a potential functional conservation of this pathway during evolution. Moreover, active novel retrogenes of NAT catalytic subunits are predominantly expressed in the mouse[116] and human testes[117] and the mouse citrate transporter homolog is also specifically expressed in the testis, the liver, and the brain[118]. A recent report identified pathogenic hNAA20 variants causing human diseases[119]. However, pleiotropic developmental defects could preclude the detection of male infertility, and tissue-specific analyses in rodents would be required to directly test the role of the mammalian NAA20 in sperm differentiation.

Our findings reveal that Nt-protein acetylation regulates male germline stem cell differentiation through proteome turnover (Fig. S7). The notion that Nt acetylation confers stability at the proteome level has been challenged recently. For example, Kats and colleagues tested the effect of all possible N-terminal di-residue combinations on the stability of a reporter protein in yeast[90]. From the 400 possible reporters only 10 were destabilised in NatB mutants, suggesting that, globally, Nt acetylation rarely controls protein abundance and turnover. Proteomic studies in yeast and human cells yielded similar results[91,92,120]. By contrast, our data identify a cell type, where NatB-mediated Nt acetylation shields key proteins from proteasomal degradation by the specific ubiquitin ligase, dUBR1. In this biological context, this is clearly the only essential function of the Nt protein acetylation. Indeed, inhibition of protein degradation fully rescues the defects in germline differentiation caused by the loss of NatB activity.

The regulatory mechanism we discovered may play an important role in sustaining germline homeostasis under fluctuating environmental conditions. In this regard, it would be interesting to extend our study to conditions of metabolic imbalance, like dietary restriction. In

plants, Nt acetylation has been studied more extensively and is known to play a critical role in stress responses. For example, it is required for pathogen tolerance[121], adaptation to osmotic stress[89] in *Arabidopsis thaliana*, and drought tolerance[122,123]. In *Drosophila*, gametogenesis is highly sensitive to the availability of dietary amino acids[9–12]. It would be of interest to investigate if a stress-mediated down-regulation of Acetyl-CoA, the co-substrate of NatB, affects both the efficacy of Nt acetylation and germline proteome turn-over.

Thus far, studies examining the influence of cell metabolism on cell fate have focused mainly on metabolic-dependent epigenetic changes affecting the transcriptome. Indeed, recent studies provide compelling evidence that changes in Acetyl-CoA production can impact transcriptional programs through histone acetylation[49–57]. Changes in cellular proteome stability due to metabolic shifts, encoded in post-translational modifications, could be a more direct and adaptable molecular mechanism for adjusting cell differentiation based on environmental conditions. Interestingly, the NatB complex displays a relatively low affinity to Acetyl-CoA, compared to HATs, with a Michaelis constant (Km) that is 50 times greater. This difference in Km values is a potential regulatory factor that could explain why, during spermatogenesis, genetically induced alterations in intracellular Acetyl-CoA level affect NatB function before impacting HAT activities. Therefore, metabolic regulation through Acetyl-CoA level alterations could signal primarily through modulation of Nt acetylation to modify protein and cell functions, a hypothesis that awaits further exploration.

## Methods
### Reagents
See Table 1 for details of antibodies, experimental organisms, oligonucleotides, software and algorithms

### Fly strains and media
Reporters: Mapmodulin$^{GFP}$ (BDSC: 51556, FlyBase ID: FBti0099819), Ncd$^{GFP}$ (BDSC: 60738, FlyBase ID: FBti0167130), Sm$^{GFP}$ (BDSC: 59815, FlyBase ID: FBti0178480), Dj$^{GFP}$ (BDSC: 5417, FlyBase ID: FBti0013334), ProtA$^{GFP}$ (gift from B. Loppin B, FlyBase ID: FBtp0023347), Gish$^{GFP}$ (BDSC: 59025, FlyBase ID: FBti0100581), Hfp$^{GFP}$ (VDRC: 318711, FlyBase ID: FBti0198685), Rbp4$^{GFP}$ (VDRC: 318563, FlyBase ID: FBti0198610), Orb2$^{GFP}$ (VDRC: 318058, FlyBase ID: FBti0198927), Loopin-1$^{GFP}$ (gift from R. Sinka, generated by[124]), Dany$^{GFP}$ (BDSC: 91773, FlyBase ID: FBti0183120), CG3927$^{GFP}$ (VDRC: 318780, FlyBase ID: FBti0198743), CG14718$^{GFP}$ (VDRC: 318741, FlyBase ID: FBti0198842), Mis12$^{GFP}$ (BDSC: 91741, FlyBase ID: FBti0214004), Vsg$^{GFP}$ (BDSC: 50812, FlyBase ID: FBti0099949), Taf1$^{GFP}$ (BDSC: 64451, FlyBase ID: FBti0181874), Mxc$^{GFP}$ (BDSC: 84130, FlyBase ID: FBti0207696), Mge$^{GFP}$ (VDRC: 318174, FlyBase ID: FBti0198764), CG7430$^{GFP}$ (VDRC: 318906, FlyBase ID: FBti0198658), Spd-2$^{GFP}$ (VDRC: 318743, FlyBase ID: FBti0198658), CG13426$^{GFP}$ (VDRC: 318517, FlyBase ID: FBti0198425), Tango5$^{GFP}$ (VDRC: 318337, FlyBase ID: FBti0198537), Vib$^{GFP}$ (BDSC: 51531, FlyBase ID: FBti0099947), CG2774$^{GFP}$ (VDRC: 318605, FlyBase ID: FBti0198336), CG5174$^{GFP}$ (BDSC: 50819, FlyBase ID: FBti0099757), Cullin 3$^{3xHA}$ (this study, see below for details), Klp10A$^{GFP}$ (BDSC: 57329, FlyBase ID: FBti0162455), Cdc42$^{GFP}$ (VDRC: 318151, FlyBase ID: FBti0198614), Vps26$^{GFP}$ (BDSC: 67153, FlyBase ID: FBti0181540).

Gal4 drivers: *bam-Gal4* (gift from M. Amoyel, FlyBase ID: FBtp0111994), *nanos-Gal4* (BDSC: 32563, FlyBase ID: FBtp0001612), *topi-Gal4* (BDSC: 91776, FlyBase ID: FBti0213638), *tj$^{NP1624}$-Gal4* (DGGR: 104055, FlyBase ID: FBti0034540), *TubP-Gal4* (BDSC: 30030, FlyBase ID: FBti0012687).

UAS transgenes: *UASt-CG7309* (this study, see below for details), *UASp-dACLY* (this study, see below for details), *UASp-dACLY$^{H772>A}$* (this study, see below for details), *UASp-dACLY$^{R380>A}$* (this study, see below for details), *UASp-dACLY$^{D1038>A}$* (this study, see below for details), *UASt-hACLY* (BDSC: 65837, FlyBase ID: FBti0183265), *UASp-dNAA25* (this study, see below for details), *UASt-dNAA20* (this study, see below for details), *UASp-dNAA20* (this study, see below for details), *UASp-*

*dNAA20$^{ERY}$* (this study, see below for details), *UASp-dNAA20$^H$* (this study, see below for details), *UASp-dNAA20$^{RG}$* (this study, see below for details), *UASp-dNAA20$^{FN}$* (this study, see below for details), *UASp-hNAA20* (this study, see below for details), *UASp-CG31851* (this study, see below for details), *UASp-CG31730$^{3xHA}$* (this study, see below for details), *UAS-Flp* (BDSC: 4539, FlyBase ID: FBti0012284), *UAS-dicer2* (VDRC#60010), *UASp-CG14740$^{3xHA}$* (this study, see below for details), *UASt-Met.GFP* (gift from Christian Klämbt, FlyBase ID: FBti0200441), *UASt-Asn.GFP* (gift from Christian Klämbt, FlyBase ID: FBti0200442), *UASp-Citron* (this study, see below for details).

RNAi transgenes: *UAS-dCS$^{RNAi}$* (BDSC: 36740, FlyBase ID: FBti0146753), *UAS-dCS$^{RNAi}$* (VDRC: 107642, FlyBase ID: FBti0120690), *UAS-dCS$^{RNAi}$* (VDRC: 26301, FlyBase ID: FBti0080130), *UAS-CG14740$^{RNAi}$* (BDSC: 60900, FlyBase ID: FBti0179283), *UAS-CG14740$^{RNAi}$* (BDSC: 31563, FlyBase ID: FBti0130599), *UAS-mAcon1$^{RNAi}$* (BDSC: 34028, FlyBase ID: FBti0140697), *UAS-mAcon1$^{RNAi}$* (VDRC: 103809, FlyBase ID: FBti0116727), *UAS-mAcon2$^{RNAi}$* (BDSC: 58074, FlyBase ID: FBti0164392), *UAS-Irp-1B$^{RNAi}$* (BDSC: 67939, FlyBase ID: FBti0186731), *UAS-Irp-1B$^{RNAi}$* (VDRC: 110637, FlyBase ID: FBti0142187), *UAS-Irp-1A$^{RNAi}$* (BDSC: 58117, FlyBase ID: FBti0164459), *UAS-Irp-1A$^{RNAi}$* (VDRC: 330238, FlyBase ID: FBti0185955), *UAS-Idh$^{RNAi}$* (BDSC: 41708, FlyBase ID: FBti0149904), *UAS-Idh$^{RNAi}$* (VDRC: 100554, FlyBase ID: FBti0120466), *UAS-Idh3a$^{RNAi}$* (VDRC: 106091, FlyBase ID: FBti0120806), *UAS-Idh3b$^{RNAi}$* (BDSC: 44475, FlyBase ID: FBti0157339), *UAS-CG32026$^{RNAi}$* (BDSC: 53953, FlyBase ID: FBti0158340), *UAS-CG3483$^{RNAi}$* (VDRC: 101958, FlyBase ID: FBti0122320), *UAS-CG5028$^{RNAi}$* (VDRC: 103834, FlyBase ID: FBti0117637), *UAS-Nc73EF$^{RNAi}$* (BDSC: 33686, FlyBase ID: FBti0140273), *UAS-CG33791$^{RNAi}$* (BDSC: 34101, FlyBase ID: FBti0140705), *UAS-CG5214$^{RNAi}$* (BDSC: 50650, FlyBase ID: FBti0157507), *UAS-ScsβG$^{RNAi}$* (BDSC: 50939, FlyBase ID: FBti0158111), *UAS-ScsβG$^{RNAi}$* (VDRC: 101554, FlyBase ID: FBti0121565), *UAS-Scsα1$^{RNAi}$* (VDRC: 107164, FlyBase ID: FBti0117489), *UAS-ScsβA$^{RNAi}$* (BDSC: 55168, FlyBase ID: FBti0159380), *UAS-ScsβA$^{RNAi}$* (VDRC: 105350, FlyBase ID: FBti0116796), *UAS-Scsα2$^{RNAi}$* (BDSC: 64025, FlyBase ID: FBti0180460), *UAS-SdhD$^{RNAi}$* (BDSC: 65040, FlyBase ID: FBti0184127), *UAS-SdhD$^{RNAi}$* (VDRC: 101739, FlyBase ID: FBti0121002), *UAS-SdhA$^{RNAi}$* (VDRC: 110440, FlyBase ID: FBti0141572), *UAS-SdhA$^{RNAi}$* (VDRC: 330053, FlyBase ID: FBti0185706), *UAS-SdhC$^{RNAi}$* (BDSC: 53281, FlyBase ID: FBti0157889), *UAS-SdhC$^{RNAi}$* (VDRC: 330697, FlyBase ID: FBti0202510), *UAS-SdhBL$^{RNAi}$* (BDSC: 58100, FlyBase ID: FBti0164431), *UAS-CG6629$^{RNAi}$* (VDRC: 106108, FlyBase ID: FBti0122779), *UAS-Fum1$^{RNAi}$* (BDSC: 51779, FlyBase ID: FBti0157741), *UAS-Fum1$^{RNAi}$* (VDRC: 105680, FlyBase ID: FBti0120862), *UAS-Fum2$^{RNAi}$* (BDSC: 77156, FlyBase ID: FBti0196089), *UAS-Fum2$^{RNAi}$* (VDRC: 106419, FlyBase ID: FBti0123418), *UAS-Fum3$^{RNAi}$* (BDSC: 67379, FlyBase ID: FBti0185631), *UAS-Fum3$^{RNAi}$* (VDRC: 103522, FlyBase ID: FBti0123817), *UAS-Fum4$^{RNAi}$* (BDSC: 65195, FlyBase ID: FBti0184282), *UAS-Fum4$^{RNAi}$* (VDRC: 103989, FlyBase ID: FBti0122623), *UAS-Mdh1$^{RNAi}$* (VDRC: 110604, FlyBase ID: FBti0142298), *UAS-Mdh2$^{RNAi}$* (BDSC: 36606, FlyBase ID: FBti0146482), *UAS-Mdh2$^{RNAi}$* (BDSC: 62230, FlyBase ID: FBti0179012), *UAS-Mdh2$^{RNAi}$* (VDRC: 101551, FlyBase ID: FBti0121546), *UAS-CG10748$^{RNAi}$* (BDSC: 62228, FlyBase ID: FBti0179010), *UAS-CG10749$^{RNAi}$* (BDSC: 62229, FlyBase ID: FBti0179011), *UAS-dACC$^{RNAi}$* (VDRC: 8105, FlyBase ID: FBti0090448), *UAS-dFASN1$^{RNAi}$* (BDSC: 28930, FlyBase ID: FBti0127757), *UAS-dFASN2$^{RNAi}$* (VDRC: 105855, FlyBase ID: FBti0119829), *UAS-dFASN3$^{RNAi}$* (BDSC: 63026, FlyBase ID: FBti0180103), *UAS-beg$^{RNAi}$* (VDRC: 108556, FlyBase ID: FBti0116633), *UAS-CG12170$^{RNAi}$* (BDSC: 40867, FlyBase ID: FBti0149775), *UAS-CG3603$^{RNAi}$* (VDRC: 107046, FlyBase ID: FBti0117232), *UAS-CG16935$^{RNAi}$* (BDSC: 36671, FlyBase ID: FBti0146682), *UAS-CG16935$^{RNAi}$* (BDSC: 43297, FlyBase ID: FBti0151309), *UAS-yip2$^{RNAi}$* (BDSC: 36874, FlyBase ID: FBti0146565), *UAS-yip2$^{RNAi}$* (VDRC: 26562, FlyBase ID: FBti0080546), *UAS-Mtpα$^{RNAi}$* (BDSC: 32873, FlyBase ID: FBti0140375), *UAS-Echs1$^{RNAi}$* (BDSC: 62221, FlyBase ID: FBti0179003), *UAS-Ppt1$^{RNAi}$* (BDSC: 62291, FlyBase ID: FBti0179684), *UAS-Ppt2$^{RNAi}$* (BDSC: 28362, FlyBase ID: FBti0127136), *UAS-Elo68beta$^{RNAi}$* (BDSC: 50646, FlyBase ID: FBti0157502),

**Table 1 | Reagents and resources**

| Reagent or resource | Source | Identifier |
|---|---|---|
| Antibodies | | |
| Chicken anti-GFP, 1/10 000 | Abcam | Cat#ab13970; RRID: AB_300798 |
| Mouse anti-pan polyglycylated Tubulin (1/5000) | Millipore | Cat#MABS276 |
| Rabbit anti-cleaved *Drosophila* Dcp-1 (Asp216), 1/500 | Cell Signaling Technology | Cat#9578 S; RRID: AB_2721060 |
| Rat anti-HA, 1/250 | Roche | Cat#11867423001; RRID: AB_390918 |
| Mouse anti-FLAG M2, 1/500 | Millipore | Cat#F3165-2MG; RRID: AB_259529 |
| Rabbit anti-*Drosophila* ICE (drICE), 1/1000 | Cell Signaling Technology | Cat#13085 S; RRID: AB_2798115 |
| Rabbit, anti-Ubiquitin (linkage-specific K48), 1/500 | Abcam | Cat#ab140601; RRID: AB_2783797 |
| Mouse anti-Ubiquitin (FK2), 1/1000 | Millipore | Cat#ST1200; RRID: AB_2043482 |
| Mouse anti-α-Tubulin, 1/1000 | Sigma-Aldrich | Cat#T6199; RRID: AB_477583 |
| Mouse anti-ATP5A, 1/500 | Abcam | Cat#ab14748; RRID: AB_301447 |
| Rabbit anti-Acetyl-Histone H3, 1/1000 | Millipore | Cat#06-599; RRID: AB_2115283 |
| Guinea pig anti-Scotti, 1/250 | PMID: 20643358 | RRID: AB_2568236 |
| Experimental models: organisms/strains | | |
| *D. melanogaster* lines – See Table S4 | Various | N/A |
| Oligonucleotides | | |
| RT-qPCR primers – See Table S3 | This paper | N/A |
| Software and algorithms | | |
| Fiji | PMID: 22743772 | https://fiji.sc/ |
| Adobe Illustrator CC 2018 | Adobe.com | N/A |
| Prism 7 GraphPad | GraphPad Software | https://www.graphpad.com/scientific-software/prism/ |

*UAS-CG17821^RNAi* (BDSC: 50898, FlyBase ID: FBti0157389), *UAS-Elo68alpha^RNAi* (BDSC: 53307, FlyBase ID: FBti0157915), *UAS-CG18609^RNAi* (BDSC: 44510, FlyBase ID: FBti0157416), *UAS-Mtpβ^RNAi* (BDSC: 34546, FlyBase ID: FBti0140715), *UAS-Hat1^RNAi* (BDSC: 42488, FlyBase ID: FBti0150967), *UAS-CG1894^RNAi* (BDSC: 34925, FlyBase ID: FBti0144900), *UAS-Ing3^RNAi* (VDRC: 109799, FlyBase ID: FBti0142087), *UAS-Ing5^RNAi* (VDRC: 102002, FlyBase ID: FBti0121514), *UAS-pont^RNAi* (BDSC: 50972, FlyBase ID: FBti0158154), *UAS-Eaf6^RNAi* (BDSC: 50518, FlyBase ID: FBti0157168), *UAS-e(y)3^RNAi* (BDSC: 32346, FlyBase ID: FBti0132041), *UAS-d4^RNAi* (BDSC: 43186, FlyBase ID: FBti0150869), *UAS-Ada2b^RNAi* (BDSC: 35334, FlyBase ID: FBti0144328), *UAS-dTat^RNAi* (BDSC: 28777, FlyBase ID: FBti0127341), *UAS-CG17003^RNAi* (VDRC: 101273, FlyBase ID: FBti0119276), *UAS-dNAA20^shRNA* (BDSC: 36899, FlyBase ID: FBti0146623), *UAS-dNAA20^RNAi* (VDRC: 109664, FlyBase ID: FBti0141766), *UAS-dNAA25^RNAi* (VDRC: 21960, FlyBase ID: FBti0080678), *UAS-dNAA25^RNAi* (VDRC: 103558, FlyBase ID: FBti0116761), *UAS-CG31851^RNAi* (VDRC: 104306, FlyBase ID: FBti0120394), *UAS-CG31730^RNAi* (VDRC: 104274, FlyBase ID: FBti0119985), *UAS-CG31730^RNAi* (VDRC: 21408, FlyBase ID: FBti0079042), *UAS-CG31730^shRNA* (BDSC: 42848, FlyBase ID: FBti0151179), *UAS-dNAA30^RNAi* (VDRC: 101769, FlyBase ID: FBti0121882), *UAS-dNAA35^RNAi* (VDRC: 109595, FlyBase ID: FBti0141310), *UAS-dNAA38^RNAi* (VDRC: 34750, FlyBase ID: FBti0080192), *UAS-CG32319^RNAi* (VDRC: 24728, FlyBase ID: FBti0079776), *UAS-CG10932^RNAi* (BDSC: 51785, FlyBase ID: FBti0157747), *UAS-CG9149^RNAi* (BDSC: 56858, FlyBase ID: FBti0163209), *UAS-CG9149^RNAi* (BDSC: 67208, FlyBase ID: FBti0185454), *UAS-Hmgs^RNAi* (BDSC: 57738, FlyBase ID: FBti0164187), *UAS-Hmgcl^RNAi* (BDSC: 51861, FlyBase ID: FBti0157828), *UAS-SCOT^RNAi* (BDSC: 51899, FlyBase ID: FBti0157866), *UAS-sro^RNAi* (BDSC: 67767, FlyBase ID: FBti0186784), *UAS-CG13377^RNAi* (BDSC: 65215, FlyBase ID: FBti0184141), *UAS-dACLY^shRNA* (BDSC: 65175, FlyBase ID: FBti0184262), *UAS-dACLY^RNAi* (VDRC: 30282, FlyBase ID: FBti0090534), *UAS-CG33934^RNAi* (VDRC: 50700, FlyBase ID: FBti0087972), *UAS-CG33934^RNAi* (VDRC: 50699, FlyBase ID: FBti0087971), *UAS-CG33934^shRNA* (BDSC: 44093, FlyBase ID: FBti0158694), *UAS-Indy-2^shRNA* (BDSC: 34891, FlyBase ID: FBti0144864), *UAS-Indy-2^RNAi* (VDRC: 51048, FlyBase ID: FBti0159828), *UAS-Indy-2^RNAi* (VDRC: 50694, FlyBase ID:

FBti0087970), *UAS-CG7309^RNAi* (VDRC: 100142, FlyBase ID: FBti0118895), *UAS-dCIC/sea^shRNA* (BDSC: 34685, FlyBase ID: FBti0140854), *UAS-dCIC/sea^shRNA* (BDSC: 33976, FlyBase ID: FBti0140637), *UAS-dUBR1^RNAi* (BDSC: 31374, FlyBase ID: FBti0130788), *UAS-dUBR1^RNAi* (VDRC: 108902, FlyBase ID: FBti0160098), *UAS-Pdha^RNAi* (BDSC: 55345, FlyBase ID: FBti0159564), *UAS-AcCoAS^RNAi* (BDSC: 41917, FlyBase ID: FBti0149942), *UAS-Acat1^RNAi* (BDSC: 51785, FlyBase ID: FBti0157747), *UAS-Acat2^RNAi* (BDSC: 56858, FlyBase ID: FBti0163209), *UAS-dUBR4^shRNA* (BDSC: 32945, FlyBase ID: FBti0140453), *UAS-dUBR5^shRNA* (BDSC: 32352, FlyBase ID: FBti0132047).

Mutants: *CG31851^KO* (this study, see below for details), *CG31851^3xHA,KI* (this study, see below for details), *dNAA20^KO* (this study, see below for details), *dNAA20^FLAG KI* (this study, see below for details), *dNAA20^FRT* (this study, see below for details), *CG14740^KO* (this study, see below for details), *Df(2 L)BSC768* (*CG31851* deficiency, BDSC: 26865, FlyBase ID: FBab0045835), *Df(3 R)Exel7312* (CG14740 deficiency, BDSC: 7966, FlyBase ID: FBab0038304), *topi > dACLY^3xHA* (this study, see below for details), *topi > dNAA25^3xHA* (this study, see below for details), *topi > dNAA20* (this study, see below for details).

Animals were reared on fly food containing (per liter): 10 g of agar, 83 g corn flour, 60 g white sugar, 34 g dry yeast and 3,75 g Moldex (per liter, diluted in ethanol). All experimental flies were kept in incubators at 25 °C or 29 °C, and on a 12 hr light/dark cycle. Flies were transferred to fresh vials every 3 days, and fly density was kept to a maximum of 15 flies per vial. For testes immunostainings, adult males were aged for 5 days before dissection. For fatty acids feeding, flies were raised on normal fly food supplemented with 0.5% oleic and arachidonic acids.

## Fertility tests

For fertility experiments, males were collected and aged for 3 days. They were then mated over five days to CantonS females (1 male with 5 females per vial). Flies were then removed and progeny was counted.

## Generation of the *CG14740* CRISPR null mutant

To generate a *CG14740* null mutant, two gRNAs targeting the *CG14740* coding sequence (gRNA 1: AGTGTTAATAGCGTGATTGGAGG, and

gRNA 2: CTTATATTCCAGGTCATTCCCGG) were cloned into the pCFD5 vector (Addgene: Plasmid #73914, generated by ref. [125]). A 1.104 kb homology arm flanking the cleavage site 1 was PCR-amplified from genomic DNA using the Q5 high-fidelity polymerase from New England Biolabs (M0491S) and the following primers: 5′-AAAAGCTA GCTGGACAAAATCAGAACGGCA-3′ and 5′-AAAACCGCGGATCACGCT ATTAACACTGATC-3′. The PCR product was digested with NheI and SacII prior to cloning into the pDsRedattP vector (Addgene: Plasmid #51019, generated by[126]). A 1.186 kb homology arm flanking the cleavage site 2 was PCR-amplified from genomic DNA using the following primers: 5′-AAAACCTAGGTCCCGGCTACGGACACGCTG-3′and 5′-AAA ACTCGAGACATGGAAGTGGAAAGGGGT-3′. The PCR product was digested with AvrII and XhoI prior to cloning into the pDsRedattP vector, containing the first homology arm. The constructs were sequence-verified and a mutant line was established through injection (Bestgene) of the 2 generated vectors (pCFD5 gRNAs and pDsRedattP homology arms) in *yw;nos-Cas9* (FlyBase ID: FBti0156858, generated by[127]) embryos. The generated deletion removed 1016 nucleotides (nt) of the *CG14740* coding sequence and replaced it with an attP landing site and a loxP-flanked 3xP3-DsRed marker.

### Generation of the d*NAA20* CRISPR null mutant
To generate a d*NAA20* null mutant, two gRNAs targeting the d*NAA20* coding sequence (gRNA 1: AAGTGGGTCAAAGTTTCTGGCGG, and gRNA 2: GAAATAATGCACGCAAATACTGG) were cloned into the pCFD5 vector. A 0.962 kb homology arm flanking the cleavage site 1, and a 0.837 homology arm flanking the cleavage site 2 were cloned by gene synthesis (Genscript) into the pDsRedattP vector using the EcoRI, XhoI restriction sites. A mutant line was established through injection (Bestgene) of the 2 generated vectors (pCFD5 gRNAs and pDsRedattP homology arms) in *yw;nos-Cas9* embryos. The generated deletion removed 1624 nt of the d*NAA20* coding sequence and replaced it with an attP landing site and a loxP-flanked 3xP3-DsRed marker.

### Generation of the double *CG31851, CG31730* CRISPR null mutant
To generate a double *CG31851, CG31730* null mutant, two gRNAs targeting the *CG31851*, and *CG31730* coding sequences (gRNA 1: CGGAGA AGTCATGACTCGCTGGG, and gRNA 2: TATCCCATGGGTCTGCCATC AGG) were cloned into the pCFD5 vector. A 1.387 kb homology arm flanking the cleavage site 1 was PCR-amplified from genomic DNA using the Q5 high-fidelity polymerase and the following primers: 5′-AA AAGCTAGCCCACTAGGCGACCCACTTAT-3′and 5′-AAAACCGCGGGCT GGGATATAATAGGAATT-3′. The PCR product was digested with NheI and SacII prior to cloning into the pDsRedattP vector. A 1.150 kb homology arm flanking the cleavage site 2 was PCR-amplified from genomic DNA using the following primers: 5′-AAAACCTAGGATCAGG ACCTGGAGCTATGGCAATC-3′and 5′-AAAACTCGAGCCACATCTCAC ACTTGGC-3′. The PCR product was digested with AvrII and XhoI prior to cloning into the pDsRedattP vector, containing the first homology arm. The constructs were sequence-verified and a mutant line was established through injection (Bestgene) of the 2 generated vectors (pCFD5 gRNAs and pDsRedattP homology arms) in *yw;nos-Cas9* embryos. The generated deletion removed 1805 nt including the entire *CG31851* coding sequence and most of the *CG31730* coding sequence and replaced it with an attP landing site and a loxP-flanked 3xP3-DsRed marker.

### Generation of the excisable FRT-flanked *dNAA20* knock-in allele
To generate an excisable FRT-flanked *dNAA20* knock-in allele, the *dNAA20* locus (1897 nt, sequences comprised between *MKP-4* and *CG14221* genes) was cloned into the RIV FRTnMCS1FRT white vector (DGRC: Plasmid#1333) using the EcoRI and AscI restriction sites. The construct was sequence-verified and a transgenic line was established through ΦC-31 integrase mediated transformation (Bestgene), using the amorphic allele of *dNAA20* generated by CRISPR-Cas9, in which

*dNAA20* locus has been replaced by an attP site. The generated allele rescue *dNAA20* null mutant phenotypes (viability and fertility).

### Generation of the excisable FRT-flanked FLAG-tagged *dNAA20* knock-in allele
To generate an excisable FRT-flanked FLAG-tagged *dNAA20* knock-in allele, the *dNAA20* locus (1897 nt, sequences comprised between *MKP-4* and *CG14221* genes) was cloned into the RIV FRTnMCS1FRT white vector (DGRC: Plasmid#1333) using the EcoRI and AscI restriction sites by gene synthesis (Genescript). A 3xFLAG tag was added before the stop codon. The construct was sequence-verified and a transgenic line was established through ΦC-31 integrase mediated transformation (Best-gene), using the amorphic allele of *dNAA20* generated by CRISPR-Cas9, in which *dNAA20* locus has been replaced by an attP site. The generated allele rescue *dNAA20* null mutant phenotypes (viability and fertility).

### Generation of the excisable FRT-flanked HA-tagged *CG31851* knock-in allele
To generate an excisable FRT-flanked FLAG-tagged *CG31851* knock-in allele, the *CG31851* locus (1842 nt, sequences comprised between *l(2)k05911* and *CG31730* genes) was cloned into the RIV FRTnMCS1FRT white vector (DGRC: Plasmid#1333) using the EcoRI and XhoI restriction sites by gene synthesis (Genescript). A 3xHA tag was added before the stop codon. The construct was sequence-verified and a transgenic line was established through ΦC-31 integrase mediated transformation (Bestgene), using the amorphic allele of *CG31851* generated by CRISPR-Cas9, in which *CG31851, CG31730* locus has been replaced by an attP site.

### Generation of the *UASp-dACLY*[3xHA] lines
To generate a wild type *UASp-dACLY* line, *dACLY* cDNA (FlyBase ID: FBpp0289823) was cloned by gene synthesis (Genscript) into the pUASp vector (DGRC: Plasmid#1189) between the EagI and XbaI restrictions sites. A 3xHA tag was added before the stop codon and the sequence targeted by *dACLY* shRNA (BDSC: 65175) was mutated to make the UAS line resistant to the shRNA (CACGACCTATGTA-GACCTGTA > TACAACGTACGTGGATTTATA, 8 mutations introduced without affecting the dACLY protein sequence). UAS lines were established through injection (Bestgene) using classical P-element mediated transformation. For the dACLY variants (H772 > A, R380 > A, and D1038 > A) mutations were introduced by gene synthesis (Genscript).

### Generation of the *topi > dACLY*[3xHA] line
To generate a wild type *topi > dACLY* line, *topi* promoter (1412 nt upstream of the start codon) was PCR amplified from genomic DNA, and cloned into the pUASp-dACLY vector between the AgeI and EagI restrictions sites. The construct was sequence-verified and a fly line was established through injection (Bestgene) using classical P-element mediated transformation.

### Generation of the *UASp-dNAA25*[3xHA] lines
To generate a wild type *UASp-dNAA25* line, *dNAA25* cDNA (FlyBase ID: FBcl0045079) was cloned by gene synthesis (Genscript) into the pUASp vector (DGRC: Plasmid#1189) between the KpnI and PspXI restrictions sites. A 3xHA tag was added before the stop codon. UAS lines were established through injection (Bestgene) using classical P-element mediated transformation.

### Generation of the *topi > dNAA25*[3xHA] line
To generate a wild type *topi > dNAA25* line, *topi* promoter (1412 nucleotides upstream of the start codon) was PCR amplified from genomic DNA, and cloned into the pUASp-dNAA25 vector digested by KpnI. The construct was sequence-verified and a fly line was established through injection (Bestgene) using classical P- element mediated transformation.

### Generation of the *UASp-dNAA20* lines

To generate a wild type *UASp-dNAA20* line, *dNAA20* cDNA (FlyBase ID: FBpp0074532) was cloned by gene synthesis (Genscript) into the pUASp-attB vector (DGRC: Plasmid#1358) between the EagI and BamHI restrictions sites. The construct was sequence-verified and a transgenic line was established through ΦC-31 integrase mediated transformation (Bestgene), using the VK05 (BDSC: 9725, FlyBase ID: FBti0076428) attP site line. For the dNAA20 variants, mutations were introduced by gene synthesis (Genscript).

### Generation of the *UASt-dNAA20* line

To generate a wild type *UASt-dNAA20* line, *dNAA20* cDNA (FlyBase ID: FBpp0074532) was subcloned into the pUASt-attB vector (DGRC: Plasmid#1419) digested by XhoI. The construct was sequence-verified and a transgenic line was established through ΦC-31 integrase mediated transformation (Bestgene), using the VK05 (BDSC: 9725, FlyBase ID: FBti0076428) attP site line.

### Generation of the *topi > dNAA20* line

To generate a wild type *topi > dNAA20* line, *topi* promoter (1412 nucleotides upstream of the start codon) was PCR amplified from genomic DNA, and cloned into the pUASp-dNAA20 vector digested by KpnI. The construct was sequence-verified and a fly line was established through injection (Bestgene) using classical P- element mediated transformation.

### Generation of the *UASp-hNAA20* line

To generate a wild type *UASp-hNAA20* line, *hNAA20* cDNA was cloned by gene synthesis (Genscript) into the pUASp-attB vector (DGRC: Plasmid#1358) between the KpnI and XbaI restrictions sites. The construct was sequence-verified and a transgenic line was established through ΦC-31 integrase mediated transformation (Bestgene), using the VK05 (BDSC: 9725, FlyBase ID: FBti0076428) attP site line.

### Generation of the *UASp-CG31851* line

To generate a wild type *UASp-CG31851* line, *CG31851* coding regions (FlyBase ID: FBpp0080033) were cloned by PCR from genomic DNA into the pUASp vector (DGRC: Plasmid#1189) between the KpnI and XbaI restrictions sites. UAS lines were established through injection (Bestgene) using classical P-element mediated transformation.

### Generation of the *UASp-CG14740*³ˣᴴᴬ line

To generate a wild type *UASp-CG14740* line, *CG14740* coding regions (1652 nt, FlyBase ID: FBpp0082025) were cloned by PCR from genomic DNA into the pUASp-attB-3xHA vector (DGRC: Plasmid#1358) between the KpnI and SpeI restrictions sites. The construct was sequence-verified and a transgenic line was established through ΦC-31 integrase mediated transformation (Bestgene), using the VK14 (BDSC: 9733, FlyBase ID: FBti0076436) attP site line.

### Generation of the *UASt-CG7309* line

To generate a wild type *UASt-CG7309* line, *CG7309* coding regions (1598nt, FlyBase ID: FBpp0290402) were cloned by PCR into the pUASt-attB vector (DGRC: Plasmid#1419) between the EcoRI and NotI restrictions sites. The construct was sequence-verified and a transgenic line was established through ΦC-31 integrase mediated transformation (Bestgene), using the VK15 (BDSC: 9736, FlyBase ID: FBti0076439) attP site line.

### Generation of the *UASp-CG31730*³ˣᴴᴬ line

To generate a wild type *UASp-CG31730* line, *CG31730* coding regions (498 nt, FlyBase ID: FBpp0080059) were cloned by PCR from genomic DNA into the pUASp-attB-3xHA vector (DGRC: Plasmid#1358) between the EagI and PspXI restrictions sites. The construct was sequence-verified and a transgenic line was established through ΦC-31 integrase

mediated transformation (Bestgene), using the VK05 (BDSC: 9725, FlyBase ID: FBti0076428) attP site line.

### Generation of the *Cullin 3*³ˣᴴᴬ line

To generate a *Cullin 3*³ˣᴴᴬ line, *Cullin 3* locus (4218 nt containing the testis specific-promoter, the coding regions and the 3'UTR region) were cloned by gene synthesis (Genscript) into the pUASp-attB-3xHA vector (DGRC: Plasmid#1358) between the KpnI, and XbaI restrictions sites. A 3xHA tag was added before the stop codon. The construct was sequence-verified and a transgenic line was established through ΦC-31 integrase mediated transformation (Bestgene), using the VK05 (BDSC: 9725, FlyBase ID: FBti0076428) attP site line.

### Generation of the *UASp-Citron* line

To generate a *UASp-Citron* line, *Citron* sequences (Addgene: Plasmid #134303, generated by[26]) was cloned by gene synthesis (Genscript) into the pUASp-attB vector (DGRC: Plasmid#1358) between the KpnI and EagI restrictions sites. The construct was sequence-verified and a transgenic line was established through ΦC-31 integrase mediated transformation (Bestgene), using the using the VK14 (BDSC: 9733, FlyBase ID: FBti0076436) attP site line.

### Immunohistochemistry

Testes were dissected, placed on poly-L-lysine (Sigma-Aldrich, P1524-1G) coated slides, fixed in 3.7% formaldehyde (Polyscience) in 1xPBS for 20 min at room temperature (RT) and then washed several times in 1xPBS. The testes were permeabilised with 0.3% sodium deoxycholate in PBSTX (1× PBS, 0.1% Tween 20, 0.3% Triton X-100) during 30 min. After PBSTX washing, the tested were blocked in PBSTX + 4% Horse serum (HS) during at least 1 h at RT. The primary antibodies incubation was performed in PBSTX + HS during 48 h at 4 °C. After several washes, secondary antibodies were incubated 2 h at room temperature. Fluorescent secondary antibodies (488-, 546- and 647-conjugated) were obtained from Jackson Immunoresearch. After DAPI staining (Sigma-Aldrich, D9542-5MG), dissected testes were mounted into Vectashield (Vector Labs). Fluorescence images were acquired using a Leica SP5 DS confocal microscope.

### Reverse transcription and quantitative-PCR

RNAs were extracted from 150 adult testes (or from 10 L3 larvae) using TRIzol (Invitrogen). RNAs were cleaned using RNAeasy mini Kit (Qiagen, 74-104). cDNAs were synthesised using the iScript cDNA synthesis kit (Bio-Rad, 170-8889) from 500 ng of total RNAs. Quantitative PCRs were performed by mixing cDNA samples (5 ng) with iTaq Universal SYBR® Green Supermix (Bio-Rad, 172-5124) and the relevant primers in 384-well plates. Expression abundance was calculated using a standard curve for each gene, and normalised to the expression of the *Tub* control gene. For data display purposes, the median of the expression abundance was arbitrarily set at 100% for control males, and percentage of that expression is displayed for all the tested genotypes. qPCR primer pairs used are listed in Table S3.

### Proteins extraction and western blotting

150 testes were dissected in 1× PBS solution and pooled per sample. Proteins were extracted with the following lysis solution: 6 M Urea, 150 mM NaCl, 50 mM Tris pH8, 1 mM EDTA and 100 mM NEM. After 30 minutes on ice, the samples were sonicated three times during 20 seconds, centrifuged at 13,000 rpm during 10 minutes, and the supernatants were collected.

The sample were then mixed with V/V 2xSDS, 100 mM DTT and heated at 100 °C during 3 min. Protein extracts were loaded on 4–15% Mini-PROTEAN® TGX Stain-Free™ Protein Gels (Biorad), run with Tris/Glycine/SDS buffer and transferred onto nitrocellulose membranes. After blocking, the membranes were incubated with the primary antibodies over night at 4 °C. The corresponding horseradish

peroxidase conjugated secondary antibodies were used at 1/1000 dilution and incubated with the membranes during 2 hours at room temperature. Signal detection was performed with Pierce™ ECL Western Blotting Substrate (Thermo Scientific) using Fuji Las 4000 (GE Healthcare).

## Statistics and data presentation

All statistical analyses were carried out in GraphPad Prism 7.04. Comparisons between two genotypes and/or conditions were analysed with the Mann–Whitney-Wilcoxon rank sum test. Multiple comparisons between a single control condition and different genotypes were analysed using one-way non-parametric ANOVA. These two non-parametric tests do not require the assumption of normal distributions, so no methods were used to determine whether the data met such assumptions. All graphs were generated using GraphPad Prism 7.04. All confocal and bright field images belonging to the same experiment and displayed together in our figures were acquired using the exact same settings. For visualisation purposes, level, and channel adjustments were applied using ImageJ to the confocal images shown in the figure panels (the same correction was applied to all images belonging to the same experiment), but all quantitative analyses were carried out on unadjusted raw images or maximum projections. In all figures, control datasets are displayed in blue, data related to loss-of-function experiments in green and results related to rescue experiments in red. When significant the differences between the medians of the control and the tested conditions are indicated.

## Reporting summary

Further information on research design is available in the Nature Portfolio Reporting Summary linked to this article.

## Data availability

All data is available in the main text or the supplementary materials. Materials generated for the study are available from the corresponding author on request. Source data are provided with this paper Source data are provided with this paper.

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

## Acknowledgements

We thank Marc Amoyel, Bénédicte Durand, Benjamin Loppin, Rita Sinka, Alan Jian Zhu, Christian Lehner, Christian Klämbt, and Ilona Grunwald Kadow for fly lines; the Bloomington Drosophila Stock Center for providing *Drosophila* fly lines; the iBV platforms: Baptiste Monterroso and Sameh Ben Aicha from the imaging facility, Marie-Christine Chaboissier, and Maximilian Fürthauer for comments and Anita Mencser for fly food; and all the members of the BH laboratory for fruitful discussions and comments. This work was supported by the Université Côte d'Azur, CNRS (ATIP-Avenir program), INSERM, European Research Council (ERC starting grant CellSex, Grant number: ERC-2019-STG#850934), and the French Government (National Research Agency, ANR) through the "Investments for the Future" programs LABEX SIGNALIFE ANR-11-LABX-0028-01 and IDEX UCAJedi ANR-15-IDEX-01.

## Author contributions

Conceptualisation: C.F., B.H., methodology: C.F., B.H., investigation: C.F., T.P., M.dD.S., C.H., B.H., visualisation: C.F., T.P., M.dD.S., C.H., Funding acquisition: B.H., project administration: B.H. Supervision: B.H., writing—original draft: B.H.

## Competing interests

The authors declare no competing interests.
