## [Peer Review File · Nature Communications]

Metabolic regulation of proteome stability via N-terminal acetylation controls male germline stem cell differentiation and reproductionREVIEWER COMMENTS

Reviewer #1 (Remarks to the Author):

Because cellular metabolism is closely associated with various intracellular molecular pathways as well as extracellular environment, metabolism-based mechanisms underlying cell functions are important and interesting to study. In this manuscript, the authors showed the testicular germ cell-specific functions of dACL_Y, an enzyme for acetyl CoA production from citrate, of the NatB complex containing dNAA20 and dNAA25 catalyzing N-terminal acetylation, which prevents proteasomal degradation of target proteins by the ubiquitin ligase dUBR1, in spermatid individualization. The results revealed an interesting molecular pathway involved in citrate metabolism in spermatogenesis, which is well supported by the presented data, though candidate target proteins for N-terminal acetylation involved in spermatid individualization are not identified in this study, which is disappointing. All the experiments are nicely conducted, and the results are clearly presented. Additional minor comments are listed below.

1. Please discuss a possible reason why Ac-CoA produced by the different pathways cannot compensate dACL_Y deficiency.
2. Fig.1G; As in other experiments, show ratio of fertile male after double and triple knock-down of the citrate transporters.

Reviewer #2 (Remarks to the Author):

In the manuscript by Francois et al., the authors show that dACL_Y-dependent production of Acetyl-CoA is required for the completion of *Drosophila* spermatogenesis only when dACL_Y uses external citrate (i.e. coming outside the gonads) as a precursor. They also show that resulting Acetyl-CoA molecules promote NatB-dependent N-terminal protein acetylation, which in turn is essential for spermatid differentiation. They finally claim that acetylated proteins could regulate the metabolic state of the fly proposing that circulating metabolites could control protein turnover and as a consequence cell functions.

Although the view of a metabolic regulation of *Drosophila* germline differentiation and

reproduction is very attractive and potentially well proposed, the first part of the results lacks of originality and, most importantly, denotes a poor attention of the authors on previously published articles dealing with the regulation of male meiosis and spermatogenesis in *Drosophila*.

First of all, the authors have not realized that the role of dACL_Y in *Drosophila* male meiosis and spermatogenesis has been extensively addressed (doi: 10.3390/ijms22115745; doi: 10.3390/cells9010206). These articles show that loss of dACL_Y affects several aspects of male meiotic cell division and not just spermatid differentiation. In addition most defects are due to the short supply of fatty acids that derive from dACL_Y activity.

Secondly, the authors have chosen not to consider a useful information, that is that dACL_Y and Sea (which corresponds to dmCIC (doi: 10.1093/hmg/ddp370) genetically interact (doi: 10.3389/fphys.2019.00383). This could have led the authors consider the most informative and obvious dACL_Y, sea double mutant combination to validate whether TCA cycle is dispensable for germline differentiation. In this case, they would have noticed that male meiosis in this combination is more defective than that of dACL_Y single mutant, thus ruling out that their working hypothesis that mitochondrial citrate is dispensable for male meiosis. This raise another criticism in their experimental set up: all results are based on RNAi-induced gene depletion, even if for most genes, classical mutant alleles (that could confirm RNAi-induced phenotype) are available.

Finally, their observations that the knocking down genes for the FA biosynthesis enzymes did not impact male fertility, is in sharp contrast with published results indicating that impairment of fatty acid synthesis alters cytokinesis and therefore spermatogenesis (doi: 10.1016/j.cub.2008.08.061; doi: 10.1093/hmg/ddp518). Once again, a more careful literature analysis could have avoided inappropriate and misleading conclusions.

Reviewer #3 (Remarks to the Author):

Comments to Francois et al,

In this manuscript the authors summarize their work on the role of N-terminal protein acetylation as a mediator to regulate spermatogenesis via sensing metabolic changes. The paper reflects a continuation of their work showing that gut derived citrate is required for proper spermatogenesis. They now expand their work to study the pathway that mediates this effect. Through careful genetic studies they show that the extracellular citrate does not serve as a TCA intermediate to increase energy expenditure but rather to provide sufficient cytosolic Acetyl-CoA. The high levels of cytosolic Acetyl-CoA are then used by NatB to N-terminally acetylate proteins and prevent them from UBR1 mediated degradation. The manuscript is well written and provides an exciting and novel pathway of metabolic sensing in testis.

While the conclusions are very convincing and well supported by the data, some questions remain and should be addressed or at least discussed in a revised version.

In their previous paper, they show that a testis specific reduction of Indy was sufficient to get a reduction in primary spermatocytes, elongating spermatids, and individualizing spermatids. This is not the case in this work. The authors should comment on it. It is a bit unsatisfying that the authors claim that “this novel proteostasis pathway 462 likely controls a relatively small fraction of the proteome “, but fail to identify any factor whose amount is regulated by NatB, ATPCL or UBR9 in testis. The fact that they observe a rescue of the NAA20 knockdown by a simultaneous knockdown of UBR1 strongly suggests a role in degradation it could also be interpreted by a role of K48 polyubiquitination in disturbing functional protei-protein interactions. The heading of panels 7 and 8 of figure 7 are in my opinion a bit misleading as they do not show protein degradation.

In this regard the authors may also discuss the publication of the Cenci lab (PMID: 31947614) suggesting that the observed phenotype of a testis specific ATPCL knockdown is at least partially caused by a reduction of fatty acids. The fact that knockdown of enzymes involved in FA synthesis do not show a sterility phenotype is intriguing but may also be due to redundancy.

Response to Reviewers

We thank the Reviewers for their time examining our manuscript and providing valuable comments. We have followed all their suggestions and believe that the new experiments and their outcome have improved our manuscript. We are providing a point-by-point response to their specific points below. Before doing so, we also highlight the critical new data included in the revised manuscript:

- 1- As requested by Reviewer #2, we performed additional experiments to assess a possible genetic interaction between *dCIC* and *dACLY*. We also realised fatty acid supplementation experiments to rule out a role of lipid biosynthesis in the *dACLY* knock-down phenotype.
- 2- We created a new genetically encoded citrate sensor to confirm that mitochondrial citrate is unnecessary for male germline differentiation.
- 3- We used new reporters to show that dUBR1-mediated K48 poly-ubiquitination triggers dUBR1 targets to the proteasome for degradation.

We present these new data in 11 new Supplementary Figure panels in a significantly revised manuscript. Changes in the manuscript are highlighted in blue.

Reviewer #1:

Reviewer #1 (Remarks to the Author):

Because cellular metabolism is closely associated with various intracellular molecular pathways as well as extracellular environment, metabolism-based mechanisms underlying cell functions are important and interesting to study. In this manuscript, the authors showed the testicular germ cell-specific functions of *dACLY*, an enzyme for acetyl CoA production from citrate, of the NatB complex containing *dNAA20* and *dNAA25* catalyzing N-terminal acetylation, which prevents proteasomal degradation of targets proteins by the ubiquitin ligase *dUBR1*, in spermatid individualization. The results revealed an interesting molecular pathway involved in citrate metabolism in spermatogenesis, which is well supported by the presented data, though candidate target proteins for N-terminal acetylation involved in spermatid individualization are not identified in this study, which is disappointing. All the experiments are nicely conducted, and the results are clearly presented. Addition minor comments are listed below.

1. Please discuss a possible reason why Ac-CoA produced by the different pathways cannot compensate *dACLY* deficiency.

This is really an interesting point. We try to clarify it (**please see page 6, lines 239-243**). One possible explanation for the lack of compensation between different pools of Ac-CoA could be due to subcellular compartmentalisation. For instance, enzymes such as *dACLY*, *AcCoAS*, and *Hmgcl* are located in the cytoplasm, while *Yip2*, *Acat1*, and *Mtpβ* act in the mitochondria, and *Acat2* is in the peroxisome. Another crucial factor to consider is the relative expression level of these Ac-CoA-producing enzymes. Interestingly, *dACLY*, among the cytosolic enzymes, has the highest expression level. According to data published by Shi (PMID: 32122991), from single-cyst staged transcriptome analyses of *Drosophila* male germline stem cell lineage, at later stages, *dACLY* is expressed twenty-four times more than *AcCoAS* and four times more than *Hmgcl*.

2. Fig.1G; As in other experiments, show ratio of fertile male after double and triple knock- down of the citrate transporters.

These data have been added and are now presented in the Fig. S1 (Fig. S11).

Reviewer #2:

In the manuscript by Francois et al., the authors show that dACL_Y-dependent production of Acetyl-CoA is required for the completion of *Drosophila* spermatogenesis only when dACL_Y uses external citrate (i.e. coming outside the gonads) as a precursor. They also show that resulting Acetyl-CoA molecules promote NatB-dependent N-terminal protein acetylation, which in turn is essential for spermatid differentiation. They finally claim that acetylated proteins could regulate the metabolic state of the fly proposing that circulating metabolites could control protein turnover and as a consequence cell functions.

Although the view of a metabolic regulation of *Drosophila* germline differentiation and reproduction is very attractive and potentially well proposed, the first part of the results lacks of originality and, most importantly, denotes a poor attention of the authors on previously published articles dealing with the regulation of male meiosis and spermatogenesis in *Drosophila*.

First of all, the authors have not realized that the role of dACL_Y in *Drosophila* male meiosis and spermatogenesis has been extensively addressed (doi: 10.3390/ijms22115745; doi: 10.3390/cells9010206). These articles show that loss of dACL_Y affects several aspects of male meiotic cell division and not just spermatid differentiation. In addition, most defects are due to the short supply of fatty acids that derive from dACL_Y activity.

We completely agree with the Reviewer's comment that we should have included a thorough discussion of previous studies in our initial manuscript. We appreciate the Reviewer for bringing this to our attention, and we want to take this opportunity to address this oversight.

Firstly, we would like to highlight a critical difference between the papers mentioned in the Reviewer's comment and our study. The previous studies utilised hypomorph alleles of *dACL_Y* that altered dACL_Y function everywhere, affecting all somatic tissues and the germline. As a result, most individuals carrying these alleles died before adult stages. Therefore, these papers investigated a developmental, irreversible phenotype of somatic origin (germline defects do not induce lethality), and it is not easy to conclude regarding the intrinsic function of *dACL_Y* in the germline in this context. The phenotype described could be caused indirectly by the dysfunctions of all the dying somatic tissues. In contrast, we were interested in the cell-autonomous functions of the citrate in the male germline, and we performed cell type-specific genetic manipulations in adult testes. We believe this crucial difference explains the distinct phenotypes we are reporting. In this context, it is not surprising that testes from ubiquitous *dACL_Y* mutation have a worse phenotype than adult testes with germline-specific *dACL_Y* knock-down.

We perform additional experiments to investigate whether lipid synthesis plays a role in the *dACL_Y* knock-down phenotype. Previous research by Di Giorgio and colleagues suggested that fatty acid supplementation could partially rescue the *dACL_Y* hypomorph phenotype. We replicated this experiment to determine if a reduction in fatty acids could also be responsible for the *dACL_Y* germline-specific knock-down phenotype. However, our results showed that fatty acid supplementation did not restore sperm individualisation or mature sperm presence in the seminal vesicles. Even though males were given fatty acids, they remained sterile. Thus, our findings indicate that the observed defects following *dACL_Y* germline-specific knock-down are not primarily due to a decrease in the supply of fatty acids.

These experiments are included in 4 new Supplementary Figure panels (Fig. S3C-F) of the revised manuscript (please see page 6, lines 263-269).

Secondly, the authors have chosen not to consider a useful information, that is that *dACL*Y and *Sea* (which corresponds to *dmCIC* (doi: 10.1093/hmg/ddp370) genetically interact (doi: 10.3389/fphys.2019.00383). This could have led the authors consider the most informative and obvious *dACL*Y, *sea* double mutant combination to validate whether TCA cycle is dispensable for germline differentiation. In this case, they would have noticed that male meiosis in this combination is more defective than that of *dACL*Y single mutant, thus ruling out that their working hypothesis that mitochondrial citrate is dispensable for male meiosis.

This is an interesting point. To investigate the potential genetic interaction between *dACL*Y and *dCIC* in the context of germline-specific *dACL*Y knock-down, we have conducted the following additional experiments:

1- First, we studied the effects of double knock-downs of *dACL*Y and *dCIC* on spermatogenesis (**Figure to Reviewers 1**). We found that individualisation was significantly impaired, resulting in a lack of motile sperm in the seminal vesicles and waste bags, marked by an activated version of caspase-3-like effector caspase. However, we observed no defects in spermatid elongation. Interestingly, our findings matched those of a single *dACL*Y knock-down, suggesting no genetic interaction between *dACL*Y and *dCIC*.

2- By conducting a functional test, we were able to demonstrate that the knock-down phenotype of *dACL*Y was not worsened by the down-regulation of *dCIC*. In fact, we were able to successfully rescue the double knock-downs by re-expressing only *dACL*Y, which restored all the previously identified defects, including male fertility, mature sperm presence, and waste bag formation. These complete rescues were achieved using both shRNA-resistant *dACL*Y transgenes, driven by *bam-Gal4* or the *topi* promoter. These functional results effectively refute the hypothesis of a genetic interaction between *dACL*Y and *dCIC* in this context, and also the notion that mitochondrial citrate is contributing significantly to the pool of cytosolic Ac-CoA of the male germline cells.

Figure to Reviewers 1. *dACL*Y and *dCIC* do not genetically interact in the male germline.

Quantifications of (A) the percentage of fertile males, (B) the percentage of seminal vesicles with mature sperm, and (C) the number of waste bags in testes of control males, males with germline-specific *dACL*Y knock-down using *bam-Gal4*, males with double *dACL*Y *dCIC* knock-down, males with rescued *dACL*Y knock-down using a *UAS-dACL*Y or a *topi>dACL*Y transgene, and males with rescued double *dACL*Y *dCIC* knock-down using a *UAS-dACL*Y or a *topi>dACL*Y transgene. (D) Representative images (DNA: DAPI, blue; protein, green) of polyglycylated α -tubulin (polyglyTub), and cleaved Dead caspase-1 (cDcp-1) expressions in testes of the same genotypes.

Scale bars: in μm . n = number of flies tested in (A), number of seminal vesicles analysed in (B), and number of testes analysed per genotype in (C).

3- To ensure the accuracy of our findings, we decided to measure the levels of cytosolic citrate in male germline cells after inhibiting mitochondrial citrate synthesis. To achieve this, we developed a new citrate sensor that could be expressed in the male germline. Our experiments conclusively found that the knock-down of *dCIC* or the double knock-down of the two mitochondrial citrate synthases did not impact the cytosolic citrate levels in the germline cells. These results, in conjunction with the data presented in Fig. 1, effectively demonstrate that mitochondrial citrate is not necessary for male germline differentiation. These new experiments are included in 2 new Supplementary Figure panels (**Fig. S1F and 1G**) of the revised manuscript (**please see page 4, lines 140-144**).

It's worth noting that the genetic interaction between *dACLY* and *dCIC*, which was previously reported in Morciano and colleagues' 2019 study, was observed in female larval neuroblasts on chromosome breaks. This means that it was a different functional readout in an unrelated cellular context. However, based on the experiments described above, we can now confirm that there is no genetic interaction between *dACLY* and *dCIC* in the context of male germline-specific genetic manipulations.

This raise another criticism in their experimental set up: all results are based on RNAi-induced gene depletion, even if for most genes, classical mutant alleles (that could confirm RNAi-induced phenotype) are available.

We also performed experiments using null mutants. We generated the first null mutants (by CRISPR) for the following genes: 1- *CG14740* (Fig. 1B), 2- *dNAA20* (Fig. 4), 3- *CG31851* and *CG31730* (Fig. S3). Furthermore, we created FRT-flanked knock-in lines for *dNAA20* that allow tissue-specific null-mutant analyses, which is rarely done in *Drosophila*. We validated the efficacy and specificity of all the RNAi lines we used through RT-qPCR and RNAi-resistance transgenes. In addition, we developed catalytic dead rescue transgenes for metabolic enzymes like *dACLY* that created over 24 new alleles. With these newly created tools, we could confirm that the studied factors' enzymatic activity mediates the phenotypes described. This is a step forward from previous studies on *dACLY*, which did not perform such experiments. Exploring beyond analyses of null mutants was beneficial to decipher how citrate impacts inter-organ communication comprehensively. More targeted approaches helped us to pinpoint the specific cell type involved and uncovered the precise mechanism responsible for the observed phenotype.

Finally, their observations that the knocking down genes for the FA biosynthesis enzymes did not impact male fertility, is in sharp contrast with published results indicating that impairment of fatty acid synthesis alters cytokinesis and therefore spermatogenesis (doi: 10.1016/j.cub.2008.08.061; doi: 10.1093/hmg/ddp518).

This is an interesting point. The two papers cited are not in contradiction with our results. Both studies are focused on genes regulating the homeostasis of very long-chain fatty acids. These genes were not part of our screen as we targeted only the genes involved in the early steps of fatty acid biosynthesis from Acetyl-CoA. Please also note we are not claiming that fatty acid synthesis is dispensable for spermatogenesis. We use our screen's positive hits (driving male sterility) to uncover a new pathway of consumption of cytosolic Ac-CoA. But we are not using the negative hits of our screen to prove that the genes tested are dispensable. Our screen identified male sterile genes, and was not designed to detect more specific cellular and molecular defects.

Reviewer #3:

In this manuscript the authors summarize their work on the role of N-terminal protein acetylation as a mediator to regulate spermatogenesis via sensing metabolic changes. The paper reflects a continuation of their work showing that gut derived citrate is required for proper spermatogenesis. They now expand their work to study the pathway that mediates this effect. Through careful genetic studies they show that the extracellular citrate does not serve as a TCA intermediate to increase energy expenditure but rather to provide sufficient cytosolic Acetyl-CoA. The high levels of cytosolic Acetyl-CoA are then used by NatB to N-terminally acetylate proteins and prevent them from UBR1 mediated degradation. The manuscript is well written and provides an exciting and novel pathway of metabolic sensing in testis. While the conclusions are very convincing and well supported by the data, some questions remain and should be addressed or at least discussed in a revised version.

In their previous paper, they show that a testis specific reduction of *Indy* was sufficient to get a reduction in primary spermatocytes, elongating spermatids, and individualizing spermatids. This is not the case in this work. The authors should comment on it.

We have clarified this point by including additional information regarding the impact of citrate transporter knock-downs on spermatid individualisation in the germline. Our previous version had yet to address this aspect. We have conducted an experiment and found that single and double downregulations of citrate transporters did not affect waste bag numbers (as shown in **Fig. S6J and S6K, please see page 11, lines 500-501**). However, when we attempted triple loss-of-functions, we observed a notable decrease in the number of individualising spermatids (by a factor of 3), similar to the phenotype resulting from *Indy* downregulation in the somatic cells of the testes.

It is a bit unsatisfying that the authors claim that “this novel proteostasis pathway 462 likely controls a relatively small fraction of the proteome”, but fail to identify any factor whose amount is regulated by NatB, ATPCL or UBR9 in testis. The fact that they observe a rescue of the NAA20 knockdown by a simultaneous knockdown of UBR1 strongly suggests a role in degradation it could also be interpreted by a role of K48 polyubiquitination in disturbing functional protei-protein interactions. The heading of panels 7 and 8 of figure 7 are in my opinion a bit misleading as they do not show protein degradation.

We agree with the Reviewer and taking these comments into consideration, we have amended the heading of all the panels of Fig. 6 and Fig. 7 accordingly (protein degradation is now replaced by protein poly-ubiquitination).

To investigate if K48 poly-ubiquitination mediated by dUBR1 is leading to target degradation, we performed two additional experiments:

1- We used two GFP reporters, which, upon proteolytic cleavage, expose either a stabilizing amino acid (methionine, Met-GFP) or a destabilizing amino acid targeted by the NatB complex (asparagine, Asn-GFP). The absence of methionine in the second reporter allowed us to replicate a lack of N-terminal acetylation. Our results clearly show that expressing the Met-GFP reporter protein in the male germline gives us a strong and stable GFP signal. On the other hand, when we express the un-acetylated NatB target, Asn-GFP, we get a very weak GFP signal. Our findings also unequivocally reveal that *dUBR1* knockdown stabilises the Asn-GFP reporter protein, indicating that dUBR1-mediated K48 poly-ubiquitination triggers dUBR1 targets to the proteasome for degradation. These new experiments are included in 2 new Supplementary Figure panels (**Fig. S6C and S6D**) of the revised manuscript (**please see page 9, lines 410-417**).

2- To test functionally if protein degradation was driving NatB loss-of-function phenotype, we also attempted rescues by downregulating proteasome subunits. Unfortunately, the proteasome is essential and necessary for normal spermatogenesis, and individualisation, so targeting ubiquitous proteasome subunits is impossible. However,

astonishingly, we found that the knockdown of several proteasomal factors, with testis-specific expression, rescued male fertility, mature sperm presence in the seminal vesicles, and waste bag formation. It's worth noting, however, that the genes we identified are as-yet uncharacterized testis-specific proteasome subunits. We're currently working on generating a CRISPR null mutant for all of them, but fully characterizing these new factors will require a lot of work and is beyond the scope of this paper.

The above-described experiments consolidated the point that dUBR1-mediated K48 poly-ubiquitination targets proteasomal degradation of the targets.

In this regard the authors may also discuss the publication of the Cenci lab (PMID: 31947614) suggesting that the observed phenotype of a testis specific ATPCL knockdown is at least partially caused by a reduction of fatty acids. The fact that knockdown of enzymes involved in FA synthesis do not show a sterility phenotype is intriguing but may also be due to redundancy.

We perform additional experiments to investigate whether lipid synthesis plays a role in the *dACLY* knock-down phenotype. Indeed, previous research by Di Giorgio and colleagues suggested that fatty acid supplementation could partially rescue the *dACLY* hypomorph phenotype in larval testes. We replicated this experiment to determine if a reduction in fatty acids could also be responsible for the *dACLY* germline-specific knock-down. However, our results showed that fatty acid supplementation did not restore sperm individualisation or mature sperm presence in the seminal vesicles. Even though males were given fatty acids, they remained sterile. Thus, our findings indicate that the observed defects following *dACLY* germline-specific knock-down are not primarily due to a decrease in the supply of fatty acids. These experiments are included in 4 new Supplementary Figure panels (**Fig. S3C-F**), of the revised manuscript (**please see page 6, lines 263-269**).

We want to emphasize that we use our screen's positive hits (driving male sterility) to uncover a new pathway of consumption of cytosolic Ac-CoA. We are not using the negative hits of our screen to prove that all the genes tested are dispensable for spermatogenesis. We would also like to highlight a critical difference between the papers mentioned in the Reviewer's comment and our study. The previous studies utilised hypomorph alleles of *dACLY* that altered *dACLY* function everywhere, affecting all somatic tissues and the germline. As a result, most individuals carrying these alleles died before adult stages. Therefore, these papers investigated a developmental, irreversible phenotype of somatic origin (germline defects do not induce lethality), and it is not easy to conclude regarding the intrinsic function of *dACLY* in the germline in this context. The phenotype described could be caused indirectly by the dysfunctions of all the dying somatic tissues. In contrast, we were interested in the cell-autonomous functions of the citrate in the male testis, and we performed germline-specific genetic manipulations on adult testes. This crucial difference explains the distinct phenotypes we are reporting.

REVIEWERS' COMMENTS

Reviewer #1 (Remarks to the Author):

The authors claimed that 'However, all double and triple loss-of-function manipulations reduced the male fertility rate by a factor of two and three respectively (Fig. 1G).' in line 147-149, and Fig.1G shows decreased progeny in the double and triple mutants, while fertile ratios of those mutants are not significantly different from control in Fig.S1I. Please more carefully discuss this point. Another my concern was addressed properly.

Reviewer #2 (Remarks to the Author):

The authors have addressed all my criticisms. However, for the sake of clarity, dCIC should be indicated as Sea (as reported in Flybase. In contrast dCIC on Flybase does not retrieve info on the mitochondrial citrate carrier). In addition, the reference about this Drosophila carrier (Carrisi et al., 2008; Morciano et al., 2009) is still missing

Reviewer #3 (Remarks to the Author):

The authors addressed all my comments sufficiently. I particularly appreciate their efforts to show the effect of N-terminal acetylation on protein stability and the use of a citrate sensor to show changes in citrate concentration in vivo.

Response to Reviewers

We thank the Reviewers again for their time examining our manuscript and providing valuable comments. We are providing a point-by-point response to their specific remarks below.

Reviewer #1 (Remarks to the Author):

The authors claimed that ‘However, all double and triple loss-of-function manipulations reduced the male fertility rate by a factor of two and three respectively (Fig. 1G).’ in line 147-149, and Fig.1G shows decreased progeny in the double and triple mutants, while fertile ratios of those mutants are not significantly different from control in Fig.S1I. Please more carefully discuss this point. Another my concern was addressed properly.

As suggested, corrections in the text clarified this point. It is now clearly stated: *“however, while remaining fertile, males with all double and triple loss-of-function manipulations displayed a fertility rate divided by a factor of two and three, respectively.”* (see lines 147-149).

Reviewer #2 (Remarks to the Author):

The authors have addressed all my criticisms. However, for the sake of clarity, dCIC should be indicated as Sea (as reported in Flybase). In contrast dCIC on Flybase does not retrieve info on the mitochondrial citrate carrier).

The suggested modification has been made. Now dCIC is also indicated as Sea. **Lines 139-140:** *“dCIC, in flies known as scheggia (sea)”*.

In addition, the reference about this Drosophila carrier (Carrisi et al., 2008; Morciano et al., 2009) is still missing.

As recommended, these two references have been added (see line 140).

Reviewer #3 (Remarks to the Author):

The authors addressed all my comments sufficiently. I particularly appreciate their efforts to show the effect of N-terminal acetylation on protein stability and the use of a citrate sensor to show changes in citrate concentration in vivo.

We want to thank the Reviewer for the positive assessment of our work.